# Malaria Molecular Surveillance in the Peruvian Amazon with a Novel Highly Multiplexed *Plasmodium falciparum* AmpliSeq Assay

Johanna Helena Kattenberg,[a] Carlos Fernandez-Miñope,[b,c] Norbert J. van Dijk,[a*] Lidia Llacsahuanga Allcca,[b] Pieter Guetens,[a] Hugo O. Valdivia,[d] Jean-Pierre Van geertruyden,[c] Eduard Rovira-Vallbona,[a§] Pieter Monsieurs,[a] Christopher Delgado-Ratto,[b,c] Dionicia Gamboa,[b,e] Anna Rosanas-Urgell[a]

[a]Institute of Tropical Medicine Antwerp, Biomedical Sciences Department, Antwerp, Belgium
[b]Instituto de Medicina Tropical Alexander von Humboldt, Universidad Peruana Cayetano Heredia, Lima, Peru
[c]Global Health Institute, University of Antwerp, Antwerp, Belgium
[d]Department of Parasitology, U.S. Naval Medical Research Unit No. 6 (NAMRU-6), Lima, Peru
[e]Departamento de Ciencias Celulares y Moleculares, Facultad de Ciencias y Filosofía, Universidad Peruana Cayetano Heredia, Lima, Peru

**ABSTRACT** Molecular surveillance for malaria has great potential to support national malaria control programs (NMCPs). To bridge the gap between research and implementation, several applications (use cases) have been identified to align research, technology development, and public health efforts. For implementation at NMCPs, there is an urgent need for feasible and cost-effective tools. We designed a new highly multiplexed deep sequencing assay (Pf AmpliSeq), which is compatible with benchtop sequencers, that allows high-accuracy sequencing with higher coverage and lower cost than whole-genome sequencing (WGS), targeting genomic regions of interest. The novelty of the assay is its high number of targets multiplexed into one easy workflow, combining population genetic markers with 13 nearly full-length resistance genes, which is applicable for many different use cases. We provide the first proof of principle for *hrp2* and *hrp3* deletion detection using amplicon sequencing. Initial sequence data processing can be performed automatically, and subsequent variant analysis requires minimal bioinformatic skills using any tabulated data analysis program. The assay was validated using a retrospective sample collection ($n = 254$) from the Peruvian Amazon between 2003 and 2018. By combining phenotypic markers and a within-country 28-single-nucleotide-polymorphism (SNP) barcode, we were able to distinguish different lineages with multiple resistance haplotypes (in *dhfr, dhps, crt* and *mdr1*) and *hrp2* and *hrp3* deletions, which have been increasing in recent years. We found no evidence to suggest the emergence of artemisinin (ART) resistance in Peru. These findings indicate a parasite population that is under drug pressure but is susceptible to current antimalarials and demonstrate the added value of a highly multiplexed molecular tool to inform malaria strategies and surveillance systems.

**IMPORTANCE** While the power of next-generation sequencing technologies to inform and guide malaria control programs has become broadly recognized, the integration of genomic data for operational incorporation into malaria surveillance remains a challenge in most countries where malaria is endemic. The main obstacles include limited infrastructure, limited access to high-throughput sequencing facilities, and the need for local capacity to run an in-country analysis of genomes at a large-enough scale to be informative for surveillance. In addition, there is a lack of standardized laboratory protocols and automated analysis pipelines to generate reproducible and timely results useful for relevant stakeholders. With our standardized laboratory and bioinformatic workflow, malaria genetic surveillance data can be readily generated by surveillance researchers and

Address correspondence to Johanna Helena Kattenberg, ekattenberg@itg.be, or Anna Rosanas-Urgell, arosanas@itg.be.

*Present address: Norbert J. van Dijk, Department of Medical Microbiology, Amsterdam University Medical Centers, Amsterdam, the Netherlands.

§Present address: Eduard Rovira-Vallbona, Barcelona Institute for Global Health, ISGlobal, Barcelona, Spain.

The authors declare no conflict of interest.

malaria control programs in countries of endemicity, increasing ownership and ensuring timely results for informed decision- and policy-making.

KEYWORDS DNA sequencing, *Plasmodium falciparum*, drug resistance, genetic epidemiology, *hrp2* and *hrp3* deletions, malaria, surveillance studies

The Global Technical Strategy for Malaria 2016–2030 identified the use of high-quality surveillance data for decision-making as an essential pillar for malaria elimination (1). In view of the current challenges in malaria control (e.g., the spread of drug resistance, changing transmission intensity, risk of malaria importation into malaria-free areas, and rapid diagnostic test [RDT] failures), genetic epidemiology is increasingly being recognized for its potential to inform national malaria control programs (NMCPs). By aligning multiple scientific fields in support of NMCP priorities, "use cases" were specified to create scenarios where genetic surveillance can be informative for decision-making (2, 3) and can be used to develop and implement new technologies.

Molecular markers associated with drug resistance are reliable predictors of treatment responses (2, 4, 5), can provide warning for emerging resistance, guide treatment policies (6), and even replace therapeutic efficacy studies in low-malaria-transmission areas (4, 5, 7, 8). The value of these markers was demonstrated in Southeast Asia, where *Plasmodium falciparum* resistance to artemisinin (ART) was associated with mutations in the Kelch protein 13 gene (*K13*) (9–12). Resistance mutations in the same gene recently emerged in Africa (13).

In addition, monitoring population genetic markers can guide control and elimination strategies by identifying drug resistance gene flow, distinguishing imported from autochthonous cases, and estimating the source of reintroduced cases in malaria-free areas (2, 14–16). Parasite genetics can identify underlying patterns of transmission and transmission intensity (17).

The gene products of *P. falciparum hrp2* and *hrp3* are detected by RDTs, and gene deletions resulting in false-negative RDT results were first detected in Peru 10 years ago (18). Since then, *hrp2* and *hrp3* deletions have been reported in many areas (19–24), raising concerns about the viability of histidine-rich protein 2 (HRP2)-based RDTs. The WHO recommends assessing the prevalence of *hrp2* and *hrp3* mutants among malaria patients and changing case management strategies when more than 5% of infections contain deletions (25).

Multiple genetic markers are required to address the above-mentioned challenges and are most efficiently targeted in a single genomic tool (3, 26). Whole-genome sequencing (WGS) can provide the most information content per sample; however, implementing WGS for surveillance is constrained by the limited access to high-throughput sequencers and the lack of technical and bioinformatic skills in countries of endemicity. As a result, sequencing is often performed through genomic consortia, delaying the turnaround time from sample collection to genetic data and limiting its usefulness to NMCPs. In addition, obtaining high-quality WGS data from low-density infections and dried blood spots (DBSs) remains challenging. Consequently, it is rare for genomic surveillance to be used beyond research (27–29).

Targeted next-generation sequencing (NGS) tools are emerging that combine numerous targets of interest into one workflow, being more time- and cost-efficient than WGS or traditional PCR-based approaches with amplicons detected by different techniques (10, 30–33). Laboratory protocols for NGS assays are well standardized, sequencing is feasible on benchtop sequencers, and automated analysis pipelines are available (2, 6), for example, the Malaria Resistance Surveillance (MaRS) protocol (34), which targets multiple drug resistance markers. While microsatellites (MSs) have frequently been used for population genetic surveillance as they are not under evolutionary pressure, these markers are not easily translated to NGS due to the short repetitive nature of the variation. Instead, single-nucleotide polymorphism (SNP) barcodes can be successfully targeted by NGS (35–37) and are informative for connectivity between populations (38, 39) and predictions of infection origins (40, 41). The SpotMalaria platform combining drug

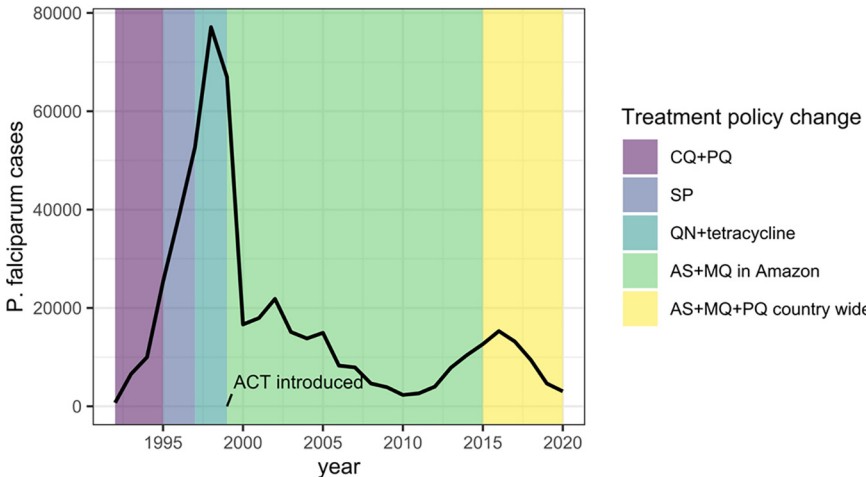

**FIG 1** Annual *P. falciparum* cases in Peru (black line) from 1992 to 2020, with first-line treatment policy changes (data from the Centro Nacional de Epidemiología, Prevención y Control de Enfermedades [2020] Sistema de Atención de Solicitudes de Acceso a la Información Pública vía Internet del Ministerio de Salud [available at http://www.minsa.gob.pe/portada/transparencia/solicitud/]). Artemisinin combination therapies (ACTs) were first introduced into the Amazon region in Peru in 1999 and the North Coast in 2001, contributing to a dramatic decline in the number of cases. The combination of artesunate, mefloquine, and primaquine (AS+MQ+PQ) was the recommended first-line treatment in the Amazon region from the start (51), with continued high efficacy (107), and the combination of AS plus sulfadoxine-pyrimethamine (SP) was used on the North Coast. In 2015, AS+MQ+PQ became the recommended first-line treatment in the whole country (108). Peru reduced the malaria case incidence by ≥40% by 2010 using passive case detection (PCD), diagnosis by light microscopy (LM), and treatment with AS+MQ as the main components of the control program (109). In 2015, with the realization that PCD missed asymptomatic infections (110), focal screening and treatment were introduced, resulting in a further reduction in the malaria incidence (50). CQ+PQ, chloroquine plus primaquine; QN, quinine.

resistance markers with a SNP barcode was helpful in tracking the rapid spread of resistant parasites in the Greater Mekong subregion (42).

Currently, there is no multifunctional tool that includes a combination of more than two types of markers (i.e., SNP barcodes and drug resistance, etc.) to serve several use cases. The characterization of *hrp2* and *hrp3* deletions still relies on PCR assays that classify deletions based on the failure to amplify targets (23). The few existing tools that combine population markers with drug resistance target short regions around validated resistance SNPs, missing the potential to detect novel resistance-associated mutations. Furthermore, many SNP panels were designed from genomes across the world and lack the resolution to study subtle patterns on a smaller geographical scale. Therefore, we designed a targeted NGS assay (Pf AmpliSeq) that combines a specifically designed barcode for the target country, 13 full-length resistance-associated genes, *hrp2* and *hrp3*, and an *ama1* microhaplotype region. The assay is applicable to DBSs and adaptable to different settings. The AmpliSeq technology (43) applies multiplex PCR to simultaneously amplify a high number of targets in a rapid procedure and allows overlapping amplicons to cover large genes.

We applied the Pf AmpliSeq assay to Peruvian samples for assay validation. *P. falciparum* in Peru makes a good case study as malaria elimination is on the regional agenda (44), with countries first targeting *P. falciparum* elimination due to its lower case load than that of *Plasmodium vivax* but also in fear of emerging ART resistance. South America has been a hot spot for chloroquine (CQ) and sulfadoxine-pyrimethamine (SP) resistance evolution (45). More recently, *K13*-mediated ART resistance emerged in Guyana (46, 47), and *K13*-independent delayed parasite clearance after ART treatment was reported in Suriname (46, 48, 49). *P. falciparum* elimination is also threatened by the increasing prevalence and spread of *hrp2* and *hrp3* deletions (21, 23).

Peru reduced its malaria case incidence by ≥40% in 2010 (Fig. 1), and after a resurgence in 2016, cases have been decreasing in the past 5 years after the introduction of focal screening and treatment (50). Past high levels of CQ and SP resistance (51–54) led

**TABLE 1** Genes of interest for drug resistance molecular surveillance of malaria in Peru

| Gene ID | Chromosome | Gene with resistance-associated SNP(s) | Drug(s) associated with resistance (reference[s])[a] |
|---------|-----------|----------------------------------------|-----------------------------------------------------|
| PF3D7_1218300 | 12 | ap2-mu | ART (68, 72, 111, 112), QN (111) |
| PF3D7_1251200 | 12 | coronin | DHA (60, 76) |
| PF3D7_0709000 | 7 | crt | CQ (113–117), PPQ (96, 118–121), AMQ (122) |
| mal_mito_3 | Mitochondrial | Cytochrome b | ATQ (123–125) |
| PF3D7_0417200 | 4 | dhfr | PYR (61, 79, 126–129), PG (130) |
| PF3D7_0810800 | 8 | dhps | SULF (61, 79, 126, 127, 129) |
| PF3D7_1362500 | 13 | Exonuclease | PPQ (131) |
| PF3D7_1343700 | 13 | K13 | ART (9, 46) |
| PF3D7_0523000 | 5 | mdr1 | CQ (132, 133); PPQ (66); MQ (65); QN, HF, ART (61, 63, 65, 66, 127, 134); AMQ (66, 122); LF (134, 135) |
| PF3D7_0112200 | 1 | mrp1 | ART, MQ, LF (66, 136) |
| PF3D7_1408000 | 14 | Plasmepsin II | PPQ (64, 131, 137) |
| PF3D7_API04900 | Apicoplast | 23S rRNA | CM (138) |
| PF3D7_0104300 | 1 | ubp-1 | ART (68, 71, 111, 112) |

[a]The list of drugs to which resistance has been observed is nonexhaustive. Validated mutations are listed in the WHO *Report on Antimalarial Drug Efficacy, Resistance and Response: 10 Years of Surveillance (2010–2019)* (7). AMQ, amodiaquine; ART, artesunate; ATQ, atovaquone; CM, clindamycin; CQ, chloroquine; DHA, dihydroartemisinin; HF, halofantrine; LF, lumefantrine; MQ, mefloquine; PG, proguanil; PPQ, piperaquine; PYR, pyrimethamine; SULF, sulfadoxine; QN, quinine.

Peru to become one of the first countries in the Americas to introduce artemisinin combination therapy (ACT) in 1999 (54–56). However, ACT efficacy and drug resistance markers have not been monitored in Peru since 2009. In 1998 to 2001, up to 20% of infections in Peru had an *hrp2* deletion, increasing to 53% with *hrp2* deletions and 37% with deletions of both genes in 2012 to 2014 (18, 19). In this study, we demonstrate the added value of molecular surveillance in Peru with different use cases using our Pf AmpliSeq assay and explore the changes in the parasite population in the past 2 decades.

## RESULTS

**Pf AmpliSeq design and performance.** A panel of 13 *P. falciparum* drug resistance-associated genes (Table 1) and corresponding variants of interest, i.e., (putative) resistance-associated mutations (see Data Set S1 in the supplemental material), were included in the assay design. The selected genes included validated mutations (4, 7) or variants relevant for Peru, i.e., those previously reported or associated with historically and currently used drugs. Additional targets were the *hrp2* and *hrp3* genes, the apical membrane antigen 1 gene (*ama1*), the conserved *Plasmodium* membrane protein gene (*cpmp*) (57), and the microsatellites (MSs) *poly-alpha*, *ARAII*, *TA81*, and *PfPK2* (58). We designed a barcode of 28 SNPs with in-country resolution in Peru, as explained in detail in the supplemental material. Primers for an AmpliSeq custom panel targeting the desired regions were designed using DesignStudio software by the Illumina concierge team. This design process makes individual amplicon testing (by PCR) before multiplexing unnecessary and resulted in two primer pools ready for library preparation. The final assay included >85% of the desired regions in 233 amplicons (Data Set S1) with various amplicon lengths (55 to 323 bp). Target regions where primer design was not possible within the constraints of the multiplex assay due to AT-rich sequences or high genetic variability were the *cpmp* region and a small section of the *crt* gene, including *crt* variant I218F.

We validated the assay with *P. falciparum* cases (*n* = 312) (Fig. 2) from multiple previous studies in Peru, laboratory strains (*n* = 5), and previously genotyped samples (*n* = 6) as controls. A high number of reads was generated in the assay (mean of 217,037 ± 490,478 paired reads/sample after trimming low-quality reads), with a median of 99.6% (range, 3.9 to 99.9%) of trimmed reads aligning to the *P. falciparum* 3D7 (Pf3D7) genome. A mean of 1,336 ± 3,627 paired reads (range, 20 to 43,795) were generated per library (including all samples and controls [also negative controls]). To improve the quality of the sequences (percentage of base calls with quality score above 30 [%$Q_{30}$]), we increased the nucleotide diversity in the run (30% GC content in the libraries) with a 20% spike-in of a PhiX library. However, while the overall %$Q_{30}$

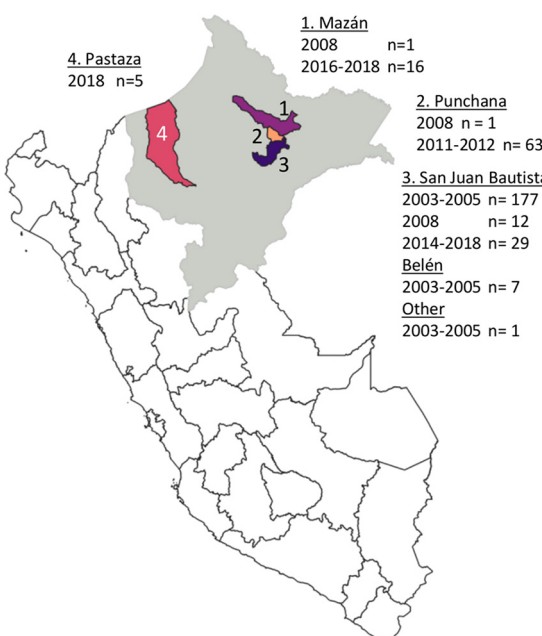

**FIG 2** Map of study sites of retrospective sample collection from the Peruvian Amazon region Loreto (gray). The majority of the samples were collected in Maynas Province (areas 1, 2, and 3): samples were collected in or near the San Juan Bautista district (area 3), which covers part of the urban area of Loreto's capital Iquitos and periurban communities south of Iquitos (*n* = 226), and the Mazán district (area 1) (*n* = 17) and the Punchana district (area 2) (*n* = 64) with remote communities living near forest rivers such as the Amazon River and the Napo River, north of Iquitos. A final collection of samples from 2018 was available from the Northern Amazon region in Datem del Marañón Province, Pastaza district (area 4) (*n* = 5). *P. falciparum*-positive samples from previous studies were selected based on geographical representation and parasite density ($\geq$100 p/$\mu$L by PCR; geometric mean density of 4,300 $\pm$ 3.1 p/$\mu$L).

increased, adding 20% PhiX is not recommended as it did not decrease the proportion of low-quality reads that were excluded (9.7% $\pm$ 10.6% of reads trimmed without a spike-in versus 10.3% $\pm$ 7.3% with a spike-in), with a 20% reduction of the total read number.

**Depth of coverage.** The median amplicon depth of coverage of aligned high-quality reads past the filter (DP) was 82.9 (interquartile range [IQR], 42.9 to 116.1) (Fig. S1). Fifteen (6.4%) amplicons had low DP values ($<$10) (Table S1), while 6 (2.6%) had high DP values ($>$150) (Table S2). Among these, 4 *hrp2* and *hrp3* amplicons had lower DP values in the study samples, but not in the controls, due to the high prevalence of deletions in these genes. The lower DP values for other amplicons are likely due to sequence variations in primer binding sites, resulting in poor amplification and large amounts of missing data for these regions, e.g., A220S in *crt*. In contrast, 2 *hrp2* amplicons had higher depths than the average, possibly due to cross-alignment between *hrp2* and *hrp3* repetitive regions. For 23/233 amplicons, no variants (i.e., only reference alleles) were detected despite good amplification and sequencing (Table S3), and these amplicons were located in conserved regions of drug resistance genes.

**Uninfected controls.** Primer specificity was tested with four uninfected human blood samples (negative controls), and none resulted in genotypes in the assay target region. We did observe background sequences with a very low number of reads corresponding to multiple unspecific hits to *Plasmodium* and human sequences; all of them were outside the assay target regions and below the filtering threshold.

Eight stored previous library preparations (including control samples and one human negative control) were included to test replicability; however, the human negative control was contaminated with *P. falciparum* sequences, probably acquired after library preparation and during storage; thus, replicability could not be assessed.

**Parasite density limit and selective whole-genome amplification.** The DP decreased while the missingness increased with decreasing parasite density in a dilution of 3D7 (6,000 to 6 parasites/$\mu$L [p/$\mu$L]) at a DNA concentration mimicking that of a DBS sample.

At the lowest density (6 p/$\mu$L), the median DP was only 10, and 60% of loci were missing. Selective whole-genome amplification (sWGA) prior to library preparation improved the mean DP and the number of high-quality reads at parasite densities of $\leq$60 p/$\mu$L (Fig. S2), with 10-fold more reads at 60 p/$\mu$L and 90-fold more reads at 6 p/$\mu$L.

**Error rate and reproducibility.** Sequencing accuracy was higher for biallelic SNPs (mean error rate of 0.008% $\pm$ 0.004%) than for indels (0.02% $\pm$ 0.005%) (Table S4). At parasite densities of >6 p/$\mu$L, sWGA is not recommended, as error rates were higher with (0.13% $\pm$ 0.06%) than without (0.05% $\pm$ 0.01%) sWGA, with little gain in coverage.

The reproducibility of the assay was tested, with a median difference of 4 SNPs (0.6% of SNP loci; 0.007% of 57,445 bp targeted in the assay) between sample and control replicates observed. With indels and multiallelic variants, a median difference of 50 alleles (2.8% of loci; 0.09% of 57,445 bp) was observed.

In comparison to previous results with laboratory strains and control samples, 97.6% of genotypes were accurate (Table S5). For the control samples, 7.9% additional minor alleles in heterozygous genotypes were detected by AmpliSeq, which were likely to be undetected in the SpotMalaria pipeline and WGS data due to lower sequencing depth.

**Complexity of infection.** Minority clones were detected by the Pf AmpliSeq assay with up to an 80:20 3D7-Dd2 mixture with 43.8% (28/64) of Dd2 loci detected (Table S6). Below a ratio of 95:5, few Dd2 genotypes were detected.

The complexity of infection (COI) estimates the number of genetically distinct clones in an infection, which can be used as a proxy for transmission intensity (16, 59). We applied different methods to determine the COI from NGS data, with results varying considerably by method (Table S7). Therefore, we determined the most frequent value (mode) of the COI, which showed an increasing proportion of multiple-clone infections in Peru over time (13.8% in 2003 to 2005, 28.8% in 2008 to 2012, and 33.3% in 2014 to 2018 [$P = 0.0005$ by a $\chi^2$ test]) (Fig. S3).

**Use case of *hrp2* and *hrp3* gene deletions.** Compared to previous PCR classifications, all samples (10/10) were correctly classified as "RDT failure" (deletion of both genes) versus "RDT detectable" (presence of one or both genes) using the read depths of *hrp2* and *hrp3* amplicons. For 6/10 samples, the classification was correct for both genes, but for 4 samples, the *hrp2* classification was inconclusive (Table S8) due to discrepancies in classifications between amplicons.

With the Pf AmpliSeq assay, 94.9% (241/254) of the study samples could be classified as RDT failure or RDT detectable. The proportion of individuals with RDT failure increased from 17.7% in 2003 to 2005 to 42.6% in 2008 to 2012 and 73.3% in 2014 to 2018 ($P > 0.001$ by a $\chi^2$ test). While *hrp3* presence versus absence could be determined for most samples (96.5%), 26.8% of *hrp3*-positive (*hrp3*$^+$) samples were inconclusive for *hrp2* (Fig. S9), but since they were not *hrp3* deleted, they were classified as RDT detectable.

**Barcode performance.** All 28 barcode SNPs were detected by the assay (Table S9), although 1 (Pf3D7_12_1127001) was genotyped as an indel. In the study samples, the population minor allele frequencies (MAFs) of the barcode SNPs ranged between 0.35 and 0.50 at 9 loci and between 0.1 and 0.34 at 12 loci and were <0.1 at 7 loci. We did not observe the minor allele of one of the barcode SNPs (Pf3D7_05_921893) in the study samples, although it was detected in laboratory strains ($n = 5$) and 5/6 control samples.

**Use case of transmission intensity: genetic diversity and differentiation.** The similarity of the samples was explored over space (district) and time (year of collection) using principal-component analysis (PCA), which showed clustering by year rather than by collection site (Fig. S4). Therefore, the samples were divided into three time periods for subsequent analyses: (i) the early time period (2003 to 2005), (ii) the intermediate time period (2008 to 2012), and (iii) the most recent period (2014 to 2018), where little variability was observed.

The genetic diversity (expected heterozygosity [$H_e$]) of the parasite population was lower in 2008 to 2012 ($P = 0.003$) and in 2014 to 2018 ($P = 0.00004$) than in 2003 to

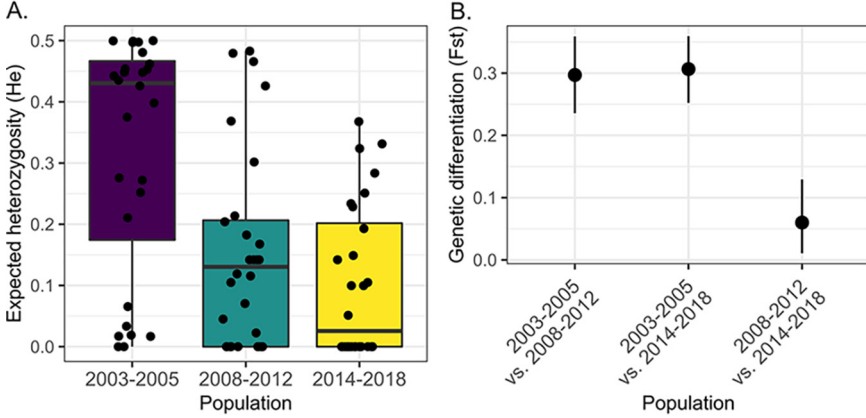

**FIG 3** Population genetic statistics. (A) Genetic diversity (expected heterozygosity) for each barcode locus (black dots) grouped by time period. Box plots show a decrease in the median (interquartile range) $H_e$ after the first time period (2003 to 2005). Since differences in $H_e$ values might be due to unequal sampling in districts over time, we assessed $H_e$ within the San Juan Bautista district, which was sampled at all three time periods. This confirmed the decreasing diversity over time, with a pronounced decrease in $H_e$ in 2014 to 2018 ($P > 0.001$) (see Fig. S5 and Table S10 in the supplemental material). (B) Genetic differentiation of the three populations measured as the $F_{ST}$ (100) using the R package diveRsity. The numbers of individuals for each population were 118 in 2003 to 2005, 65 in 2008 to 2012, and 38 in 2014 to 2018. $H_e$ values by time and district are plotted in Fig. S5, and the differentiation measures $G'_{ST}$ (101) and Jost's $D$ (102) are plotted in Fig. S6.

2005 (Fig. 3A). The total population size was small, as the observed heterozygosity ($H_{obs}$) was lower than the $H_e$ ($P < 0.005$), indicating that genetic diversity was lower than expected from a population in Hardy-Weinberg equilibrium. The parasites in 2008 to 2012 and 2014 to 2018 showed great genetic differentiation ($F_{ST} > 0.25$) compared to parasites in 2003 to 2005 (Fig. 3B). The parasite population changed considerably since 2008, resulting in a less diverse and smaller population.

**Use case for connectivity of parasite populations: barcode multilocus lineages and population structure.** Using the 28-SNP barcode, we identified 36 multilocus (ML) lineages with distinct barcodes and investigated their dynamics in the study population over time (Fig. 4) and districts (Table 2). The distinct lineages clarify the significant linkage disequilibrium (LD) in the barcode observed for all time periods and districts (Table S11). In 2003 to 2005, we observed 23 lineages, and we observed only 11 in 2008 to 2012, offering an explanation for the decrease in genetic diversity.

One lineage (155) became predominant after 2008 in coexistence with other (related) lineages within district and period (Table 2 and Fig. S7). This explains the increasing number of barcode alleles that became fixed (2 alleles in 2003 to 2005, 8 in 2008 to 2012, and 14 in 2014 to 2018) (Table S12). The ML lineages in later periods (2008 to 2018) are distinct from those observed in 2003 to 2005 (Fig. S7), paralleling the observed $F_{ST}$.

With discriminant analysis of principal components (DAPC), which is similar to PCA but maximizes the differences between predefined populations, we increased the resolution to all biallelic variants ($n_{loci} = 772$), and we were able to determine the underlying causes of the observed differentiation (Fig. 5). The highest-contributing alleles along the first axis (2003 to 2005 versus 2008 to 2018) were the drug resistance mutations *dhfr* C50R and N51I, *mdr1* D1246Y, and *dhps* K540E (Table S13). Contributors to the differentiation along the second axis (2014 to 2018 and Pastaza) were seven barcode SNPs and *dhfr*, *mdr1*, *coronin*, and *ubp1* variants.

**Use case for drug resistance: artemisinin resistance-associated genes.** The full-length *K13* gene was amplified with 12 overlapping amplicons. Although ART resistance-validated SNPs were detected in control samples, we did not observe these SNPs (F446I, N458Y, M476I, Y493H, R539T, I543T, P553L, R561H, P574L, and C580Y) in the study samples. We observed several low-frequency mutations in the K13 propellor region (V445A, I448K, G449C, V581I, and a premature stop codon at position 613)

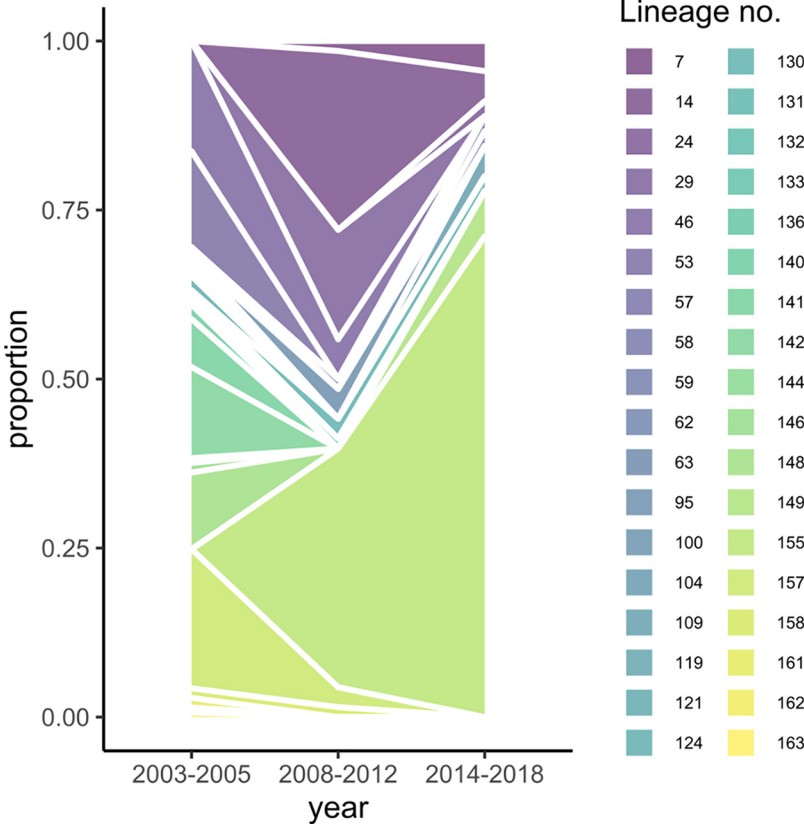

**FIG 4** Dynamics of 28-SNP-barcode multilocus lineages. An area plot shows the proportions of each ML lineage detected in the three different time periods in study samples (*n* = 254) from Peru. ML lineages were identified, clustered using the genetic distance between barcodes with the poppr package in R, and given an identification number. ML lineages differed by multiple SNPs (≥4 SNPs), enabling lineage classification even when barcodes had some missing or heterozygous genotypes. Lineages are characterized by unique barcodes and resistance genotypes and are listed in Table 2.

(Table 3), always as minor alleles in polyclonal infections. Outside the propeller region, the K189T mutation was observed at a high frequency (63% to 84%).

Forty-four amplicons covered the *ubp1* gene (91.8% of the gene), including all variants of interest. Previously reported variants (including R3138H) were not detected, but we observed a change in the predominant *ubp1* haplotype. In 2003 to 2005, a haplotype of Q107L, commonly with K1193T, was most frequently observed (42.6%) (Table 3). In later years, R1133S haplotypes (often linked with E1011K) replaced the Q107L haplotypes.

Eleven amplicons amplified the full-length *coronin* gene. We identified two novel mutations, V62M and V424I, and none of the *in vitro*-associated variants (60). V62M was seen at 20.6% in 2003 to 2005 but decreased over time (Table 3). V424I increased from 55.4% in 2003 to 2005 to 80.9% in 2014 to 2018 (Table 3). While V62M is located in the WD-40 beta-propeller domain (containing the resistance-associated mutations G50E, R100K, and E107V), V424I is located just outside this region and is not expected to contribute to ACT resistance.

**Use case for drug resistance: CQ and ACT partner drug resistance-associated genes.** The SP resistance-associated genes were each targeted with 9 amplicons, covering 100% of the *dhfr* gene and 87% of *dhps*, including all variants of interest. For 2003 to 2005, we observed predominantly single (69.5%) and double (22.7%) *dhfr* mutants and wild-type (wt) *dhps* (≥56.7%) (Table 3). Although SP use was discontinued in 1999, after 2008, SP resistance increased, with a high frequency of triple mutants in *dhfr* and *dhps*. Consistent with previous reports (45, 61), we did not detect *dhfr* C59R in Peru. Therefore, the "sextuple" mutant here is different from the African sextuple mutant, with C59R instead of C50R, but with a similar superresistance phenotype (62).

**TABLE 2** Barcode multilocus lineage characteristics[a]

| Barcode multilocus lineage | Sequence | No. of samples | | | | | | | | Variant(s) of interest for drug resistance (no. of samples) | | | | | | | Both HRP2 and HRP3 deleted (RDT failure) (no. of samples) |
| | | 2003–2005 | | | 2008–2012 | | 2014–2018 | | | | | | | | | | |
| | | Unknown district | Belen | San Juan Bautista | Punchana | San Juan Bautista | Mazan | San Juan Bautista | Pastaza | dhfr | dhps | crt | mdr1 | coronin | K13 | ubp1 category | |
|---|---|---|---|---|---|---|---|---|---|---|---|---|---|---|---|---|---|
| 53 | GGTGTCTATCTCATCGCGGGGTTCATC | 0 | 1 | 19 | 0 | 0 | 0 | 0 | 0 | Single mutant | wt | CVMNT | NDFCDD | V424I | K189T | Q107L and/or K1193T variant | No |
| 142 | GGTGTCTATTTCGGTAGGAGTTTCATC | 0 | 0 | 19 | 0 | 0 | 0 | 0 | 0 | Single mutant | Single syn mutation at position 540 | CVMNT | NDFCDD | V424I | K189T | R1133S variant | No |
| 148 | AGTGTCCGTCCAATCGCGGGATTTTTC | 0 | 0 | 16 | 0 | 0 | 0 | 0 | 0 | Double mutant | wt | SVMNT | NGFSDD | wt | wt | R1133S variant | No |
| 141 | GGCGCCCGTCCAATCGCCGGATTACTACC | 0 | 0 | 10 | 0 | 0 | 0 | 0 | 0 | Double mutant | Double mutant | SVMNT | NDFCDY | wt | K189T | R1133S + E1011K variant | No (9)/yes (1) |
| 124 | GGTGCCCGTCCCATCGCCGGATTTTTC | 0 | 0 | 3 | 0 | 0 | 0 | 0 | 0 | Double mutant | Double mutant/ triple mutant | SVMNT | NDFCDY | V424I/ wt | wt | Q107L and/or K1193T variant | No |
| 140 | GGTGTCTATCTCATCGCGGGACTTCATC | 0 | 0 | 3 | 0 | 0 | 0 | 0 | 0 | Single mutant | wt | CVMNT | NGFSDD | V424I | K189T | Q107L and/or K1193T variant | No |
| 146 | GGTXXCTATCTCATCGXGGGGCTTCATC | 0 | 0 | 2 | 0 | 0 | 0 | 0 | 0 | Single mutant | wt incomplete | CVMNT | Various | V424I | wt incomplete | wt incomplete | No |
| 161 | GGTGCCCXTCCCATCGCCGGATTATTACC | 0 | 0 | 2 | 0 | 0 | 0 | 0 | 0 | Double mutant | Double mutant/ triple mutant | SVMNT | NDFCDY | V424I/ wt | K189T | Q107L and/or K1193T variant | No |
| 62 | GGCGCCTATCTCATCGCGGGATTTACC | 0 | 0 | 1 | 0 | 0 | 0 | 0 | 0 | Single mutant | wt | CVMNT | NDFCDY | V424I | K189T | Various | No |
| 63 | GGCGCCTATCTAATXGCCGGATTATNATC | 0 | 0 | 1 | 0 | 0 | 0 | 0 | 0 | Single mutant | Incomplete mutant | CVMNT | Various | V424I | K189T | | No |
| 100 | GGGXGCTGXCCXGGXNCXGGATTANTANC | 0 | 0 | 1 | 0 | 0 | 0 | 0 | 0 | Single mutant | Incomplete mutant | CVMNT | NDFCDD | V424I | K189T | Various | No |
| 104 | GGGXGCTATCTXGGTTXGXAGTTCCACT | 0 | 0 | 1 | 0 | 0 | 0 | 0 | 0 | Single mutant | wt | CVMNT | NDFCDD | V62M | K189T | R1133S variant | No |
| 119 | GGXXXCTXTNTXNTNGXXXGTTXCXTC | 0 | 0 | 1 | 0 | 0 | 0 | 0 | 0 | Single mutant | Single mutant | CVMNT | NDFCDD | V424I | G449C | Incomplete only wt | No |
| 121 | GGXXXCCGTTXATTGXXGGATTACCACC | 0 | 0 | 1 | 0 | 0 | 0 | 0 | 0 | Double mutant | wt | SVMNT | NGFSDD | V62M | K189T | R1133S variant | No |
| 131 | GAXAXTTAXCCXGCTXGXGAACTATCATT | 0 | 0 | 1 | 0 | 0 | 0 | 0 | 0 | wt | wt | CVMNT | NDFSDD | V62M | wt incomplete | R1133S + E1011K variant | No |
| 133 | GXXGXCTAXTTXATCGNXGGGTTTCATC | 0 | 0 | 1 | 0 | 0 | 0 | 0 | 0 | Single mutant | wt | CVMNT | NDFCDD | V424I | K189T | Incomplete only wt | No |
| 136 | GXXXXCTAXTTXGTTXXXGAGXTCCAXC | 0 | 0 | 1 | 0 | 0 | 0 | 0 | 0 | | wt incomplete | CVMNT | Incomplete | V62M | wt incomplete | Incomplete only wt | Yes |
| 144 | XGTXCXTXTXXCXTCXCXGATTAXCXTC | 0 | 0 | 1 | 0 | 0 | 0 | 0 | 0 | Incomplete mutant | Double mutant | | Incomplete | wt | wt incomplete | Double mutant | No |
| 162 | XACXCXTXTXTCXGTTXGXAGCTAXCXTT | 0 | 0 | 1 | 0 | 0 | 0 | 0 | 0 | Single mutant | Single syn mutation at position 540 | | NG_SDD | V62M | | | No |
| 163 | XAXXXCNXTTXXNTTXGXAGTTXTXTC | 0 | 1 | 0 | 0 | 0 | 0 | 0 | 0 | wt | wt incomplete | | Various | V62M | wt incomplete | Incomplete only wt | No |
| 157 | GGCGCCTATCCCATCGCCGGGTTACTACC | 0 | 0 | 29 | 0 | 2 | 0 | 0 | 0 | Single mutant | wt | SVMNT | NDFCDD | V424I | K189T | Q107L and/or K1193T variant | No |
| 46 | GGTGCCTATTTAGTTGAGGAGTTACCACC | 1 | 0 | 22 | 1 | 3 | 0 | 0 | 0 | Single mutant | Single mutant/ wt | CVMNT | NGFSDD | V62M | K189T | Q107L and/or K1193T variant | Yes |
| 158 | GGCGXCCGTCCXATCGXCGXATTACTTTC | 0 | 0 | 2 | 1 | 0 | 0 | 0 | 0 | Double mutant/ triple mutant | Double mutant | SVMNT | NDFCDY | V424I/ wt | K189T | R1133S + E1011K variant | Yes (2)/no (1) |

**TABLE 2** (Continued)

| Barcode multilocus lineage | Sequence | No. of samples | | | | | | | | Variant(s) of interest for drug resistance (no. of samples) | | | | | | | Both HRP2 and HRP3 deleted (RDT failure) (no. of samples) |
|---|---|---|---|---|---|---|---|---|---|---|---|---|---|---|---|---|---|
| | | 2003–2005 | | | 2008–2012 | | 2014–2018 | | | dhfr | dhps | crt | mdr1 | coronin | K13 | ubp1 category | |
| | | Unknown district | Belen | San Juan Bautista | Punchana | San Juan Bautista | Mazan | San Juan Bautista | Pastaza | | | | | | | | |
| 155 | GGCGCCCATCTAATCGCCGGGATTACTTTC | 0 | 0 | 0 | 24 | 0 | 11 | 21 | 0 | Triple mutant | Triple mutant | SVMNT | NDFCDY | V424I (and 1 wt) | K189T/wt (1) | R1133S + E1011K variant | Yes (51)/no (4) |
| 29 | NNCGCCCATCTAATCGCCGGGATTACTTTC | 0 | 0 | 0 | 11 | 0 | 0 | 0 | 0 | Triple mutant | Triple mutant | SVMNT | NDFCDY | V424I | K189T/wt (1) | R1133S + E1011K variant | No |
| 95 | GGGCGCNATCTAATCGCCGGGATTACTTTC | 0 | 0 | 0 | 3 | 0 | 0 | 0 | 0 | Triple mutant | Triple mutant | SVMNT | NDFCDY | V424I | K189T | R1133S + E1011K variant | Yes |
| 14 | AACGTCCGTCCAATCGCGGGATTACTTCC | 0 | 0 | 0 | 16 | 2 | 0 | 2 | 0 | Triple mutant | Triple mutant | SVMNT | NDFCDY | wt | wt/K189T (2) | R1133S + E1011K variant | No |
| 130 | GACGTCCATCCAATCGCGGGATTACTTTC | 0 | 0 | 0 | 2 | 0 | 0 | 0 | 1 | Triple mutant | Triple mutant | SVMNT | NDFCDY | V424I/wt | K189T/wt | R1133S + E1011K variant | No (2)/yes (1) |
| 7 | AACGTCCATCTAATCGCGGGATTACTTTC | 0 | 0 | 0 | 1 | 0 | 0 | 2 | 0 | Triple mutant | Triple mutant | SVMNT | NDFCDY | V424I | wt/G449C | R1133S + E1011K variant | No |
| 57 | GGNGCCNATTTAATTGCGGAGTTACCACC | 0 | 0 | 0 | 0 | 1 | 0 | 0 | 0 | | Single mutant | | CVMNT/SVMNT | NGFSDD | V62M | K189T | Various |
| Yes 132 | GAXXXCCATCTXATCGCXGGATTACTTTC | 0 | 0 | 0 | 1 | 0 | 0 | 0 | 0 | Triple mutant | Triple mutant | SVMNT | NDFCDY | V424I | K189T | R1133S + E1011K variant | No |
| 149 | GGCGCCCGTCCAATTGCGGAGTTACCACC | 0 | 0 | 0 | 0 | 0 | 0 | 0 | 3 | Triple mutant | wt | SVMNT | NDFCDY | wt | K189T | Q107L and/or K1193T | No |
| 109 | GGCGCCCGTCTAATTGAGGGATTACTTTC | 0 | 0 | 0 | 0 | 0 | 0 | 2 | 0 | Triple mutant | Triple mutant | SVMNT | NDFCDY | wt | K189T (1)/wt (1) | R1133S + E1011K variant | No |
| 24 | AXXXXCCNXCCXATCNXXGGATTACTTCC | 0 | 0 | 0 | 0 | 0 | 1 | 0 | 0 | Triple mutant | Incomplete mutant | SVMNT | NDFCDY | V424I | wt incomplete | R1133S + E1011K variant | No |
| 58 | GGCGTCCATTTAATTGCCGGATTACTATC | 0 | 0 | 0 | 0 | 0 | 0 | 0 | 1 | Triple mutant | Triple mutant | SVMNT | NDFCDY | wt | K189T | R1133S + E1011K variant | No |
| 59 | GGCGNCCNTCTAATXGCNGGATTACTTNC | 0 | 0 | 0 | 0 | 0 | 0 | 1 | 0 | Triple mutant | Triple mutant | SVMNT | NDFCDY | V424I | K189T | R1133S + E1011K variant | No |

[a] ML lineages were identified, clustered using the genetic distance between barcodes with the poppr package in R, and given a number as an identifier. Heterozygote genotypes in the barcode were included, resulting in the creation of some polyclonal genotypes in barcodes, which included barcodes from other lineages. Lineages 29 and 95 are, in fact, the same as lineage 155, with additional minor clones. wt, wildtype (3D7) genotype; syn, synonymous.

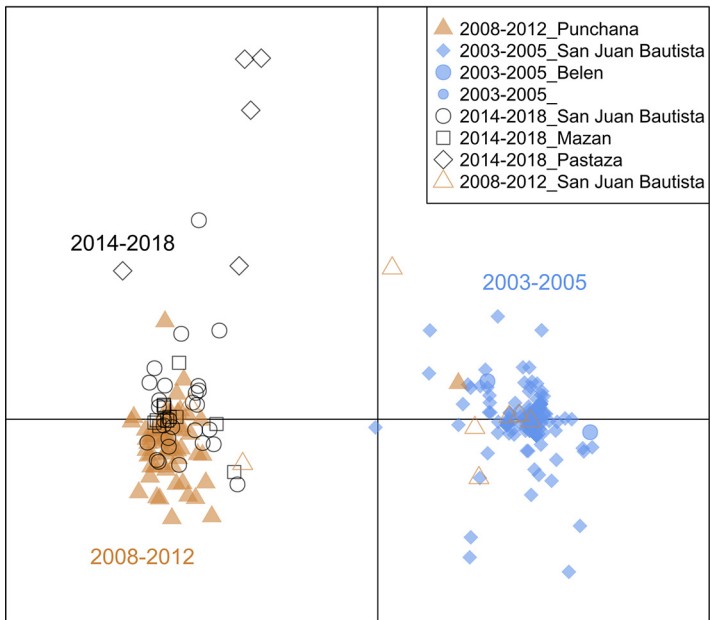

**FIG 5** Discriminant analysis of principal components (DAPC). A scatterplot of discriminant analysis eigenvalues 1 and 2 using all biallelic SNPs in the core targeted region (excluding amplicons targeting repetitive regions and nonnuclear targets) shows the differentiation between Peruvian *P. falciparum* samples collected before 2008 and those collected afterward (*x* axis) (PC1) and a subsequent separation along the *y* axis (PC2) from 2014 to more recent years and along geographic distance. DAPC calculates the discriminant components using predefined populations in such a way that samples from the same population will be grouped, while it simultaneously maximizes the distance with samples from other populations. SNPs contributing most to the DAPC are listed in Table S13 in the supplemental material. DAPC was performed with 40 principal components and 7 discriminants as determined by cross-validation. The increased resolution of the DAPC allowed the detection of greater genetic variation in samples from 2014 to 2018 (black shapes) than with the lineage analysis.

Sixteen amplicons covered the *crt* gene (91.1%), missing one variant of interest (I218F) associated with piperaquine (PPQ) resistance. The CQ resistance (CQR) *crt* haplotypes CVMNT (CQR) and SVMNT (highly CQR) increased over time (Table 3), with 100% SVMNT in 2014 to 2018. Mutation I356V, whose contribution to CQR is unclear, was frequent (>93%) in all time periods.

Eighteen amplicons covered the *mdr1* gene (87%), targeting all variants of interest associated with CQ and mefloquine (MQ) resistance (7). While the *mdr1* haplotypes NDFCDD and NGFSDD were predominant in 2003 to 2005, in later years, mainly the NDFCDY haplotype was found (Table 3). There are conflicting results on the association of NDFCDY with susceptibility to MQ, artesunate (AS), and CQ (61, 63). The *mdr1* D144G SNP has so far been reported only in Peru (annotated as D142G previously [61]) and has been decreasing in recent years.

Copy number variations (CNVs) in the *mdr1* and plasmepsin II genes associated with PPQ (64) and MQ (65) resistance, respectively, were not previously observed in Peru (66). We confirmed single copies of both genes by quantitative PCR (qPCR) in a subset of 78 samples (Fig. S8). While both genes were sequenced in our assay, the lack of increased copy numbers resulted in insufficient data to validate CNV detection by Pf AmpliSeq.

We detected 10.6% S160N mutants in 2003 to 2005 in the *ap2-mu* gene (9 amplicons covering 99.1% of the gene), which decreased in later periods (1.5% and 2.2%). No variants of interest were observed in the *pfmrp1* (95.1% covered), *P. falciparum* exonuclease (98.6% covered), cytochrome *b* (100% covered), and 23S rRNA (100% covered) genes.

**Use case for drug resistance gene flow: drug resistance and parasite lineage evolution.** Between 2003 and 2005, we observed lineages with predominantly single and double *dhfr* mutants, wt *dhps*, CVMNT and SVMNT *crt* haplotypes, multiple *mdr1* haplotypes, and the presence of *hrp2* and *hrp3* deletions (Table 2). From 2008, the

**TABLE 3** Haplotype frequencies in ART and partner drug resistance-associated genes observed in at least 1 sample[b]

| Drug(s) and gene haplotype(s) | 2003–2005 | | 2008–2012 | | 2014–2018 | | $\chi^2$ P value |
|---|---|---|---|---|---|---|---|
| | No. of samples with haplotype/ total no. of samples | % of samples with haplotype (95% CI) | No. of samples with haplotype/ total no. of samples | % of samples with haplotype (95% CI) | No. of samples with haplotype/ total no. of samples | % of samples with haplotype (95% CI) | |
| ART | | | | | | | |
| *K13* | | | | | | | |
| G449C[a] | 1/141 | 0.7 (0.1–4.9) | 0/68 | 0 | 1/45 | 2.2 (0.3–73.4) | 0.03621 |
| K189T | 94/141 | 66.7 (58.4–74.0) | 42/68 | 61.8 (49.6–73.6) | 25/45 | 55.6 (40.7–73.4) | |
| wt | 15/141 | 10.6 (6.5–17.0) | 16/68 | 23.5 (14.9–73.6) | 3/45 | 6.7 (2.1–73.4) | |
| wt incomplete | 29/141 | 20.6 (14.6–28.1) | 10/68 | 14.7 (8.0–73.6) | 14/45 | 31.1 (19.2–73.4) | |
| Missing | 2/141 | 1.4 (0.4–5.6) | 0/68 | 0 | 2/45 | 4.4 (1.1–73.4) | |
| *ubp1* | | | | | | | |
| Q107L and/or K1193T type | 60/141 | 42.6 (34.6–50.9) | 6/68 | 8.8 (4.0–18.4) | 3/45 | 6.7 (2.1–19.0) | <2.2e−16 |
| R1133S type | 21/141 | 14.9 (9.9–21.8) | 7/68 | 10.3 (4.9–20.2) | 1/45 | 2.2 (0.3–14.6) | |
| R1133S + E1011K type | 33/141 | 23.4 (17.1–31.2) | 53/68 | 77.9 (66.4–86.3) | 41/45 | 91.1 (78.3–96.7) | |
| Polyclonal Q107L/K1193T+ R1133S | 3/141 | 2.1 (0.7–6.4) | 1/68 | 1.5 (0.2–9.9) | 0/45 | 0 | |
| wt or wt incomplete | 24/141 | 17.0 (11.6–24.2) | 1/68 | 1.5 (0.2–9.9) | 0/45 | 0 | |
| *coronin* | | | | | | | |
| V424I | 78/141 | 55.3 (47.0–63.4) | 41/68 | 60.3 (48.2–71.3) | 37/45 | 82.2 (68.0–91.0) | 0.002743 |
| V62M | 29/141 | 20.6 (14.6–28.1) | 5/68 | 7.4 (3.1–16.6) | 0/45 | 0 | |
| wt | 30/141 | 21.3 (15.3–28.9) | 20/68 | 29.4 (19.7–41.4) | 8/45 | 17.8 (9.0–32.0) | |
| Polyclonal | 0/141 | 0 | 1/68 | 1.5 (0.2–9.9) | 0/45 | 0 | |
| Incomplete | 4/141 | 2.8 (1.1–7.4) | 1/68 | 1.5 (0.2–9.9) | 0/45 | 0 | |
| SP | | | | | | | |
| *dhfr* | | | | | | | |
| Single mutant (S108N) | 98/141 | 69.5 (61.4–76.6) | 6/68 | 8.8 (4.0–18.4) | 0/45 | 0 | <2.2e−16 |
| Double mutant + BR + I164L | 32/141 | 22.7 (16.5–30.4) | 0/68 | 0 | 0/45 | 0 | |
| Triple mutant | 0/141 | 0 | 61/68 | 89.7 (79.8–95.1) | 44/45 | 97.8 (85.4–99.7) | |
| Incomplete | 3/141 | 2.1 (0.7–6.4) | 0/68 | 0 | 1/45 | 2.2 (0.3–14.6) | |
| Missing | 6/141 | 4.3 (1.9–9.2) | 1/68 | 1.5 (0.2–9.9) | 0/45 | 0 | |
| *dhps* | | | | | | | |
| Single mutant K540K | 16/141 | 11.3 (7.0–17.8) | 0/68 | 0 | 0/45 | 0 | <2.2e−16 |
| Single mutant A437G | 3/141 | 2.1 (0.7–6.4) | 1/68 | 1.5 (0.2–9.9) | 0/45 | 0 | |
| Double mutant | 14/141 | 9.9 (5.9–16.1) | 0/68 | 0 | 0/45 | 0 | |
| Triple mutant | 2/141 | 1.4 (0.4–5.6) | 48/68 | 70.6 (58.6–80.3) | 30/45 | 66.7 (51.6–79.0) | |
| Incomplete mutant | 7/141 | 5.0 (2.4–10.1) | 13/68 | 19.1 (11.4–30.4) | 12/45 | 26.7 (15.7–41.6) | |
| wt | 80/141 | 56.7 (48.4–64.7) | 4/68 | 5.9 (2.2–14.8) | 2/45 | 4.4 (1.1–16.4) | |
| wt incomplete | 19/141 | 13.5 (8.7–20.2) | 2/68 | 2.9 (0.7–11.2) | 1/45 | 2.2 (0.3–14.6) | |
| CQ and MQ | | | | | | | |
| *crt* (positions 72–76) | | | | | | | |
| CVMNT | 65/141 | 46.1 (38.0–54.4) | 2/68 | 2.9 (0.7–11.2) | 0/45 | 0 | 8.37e−16 |
| SVMNT | 49/141 | 34.8 (27.3–43.0) | 60/68 | 88.2 (78.0–94.1) | 32/45 | 71.1 (56.1–82.6) | |
| CVMNT/SVMNT polyclonal | 0/141 | 0 | 1/68 | 1.5 (0.2–9.9) | 0/45 | 0 | |
| Incomplete | 27/141 | 19.1 (13.4–26.6) | 5/68 | 7.4 (3.1–16.6) | 13/45 | 28.9 (17.4 to 43.9) | |
| *mdr1* (positions 86, 144, 184, 1034, 1042, 1246) | | | | | | | |
| NDFCDD | 61/141 | 43.3 (35.3–51.6) | 1/68 | 1.5 (0.2–9.9) | 0/45 | 0 | <2.2e−16 |
| ND_CDD | 2/141 | 1.4 (0.4–5.6) | 0/68 | 0 | 0/45 | 0 | |
| NDFCDY | 17/141 | 12.1 (7.6–18.6) | 60/68 | 88.2 (78.0–94.1) | 42/45 | 93.3 (81.0–97.9) | |
| NGFSDD | 43/141 | 30.5 (23.4–38.6) | 5/68 | 7.4 (3.1–16.6) | 0/45 | 0 | |
| NG_SDD | 4/141 | 2.8 (1.1–7.4) | 0/68 | 0 | 0/45 | 0 | |
| NDFSDD | 1/141 | 0.7 (0.1–4.9) | 0/68 | 0 | 0/45 | 0 | |
| Polyclonal | 5/141 | 3.5 (1.5–8.3) | 0/68 | 0 | 0/45 | 0 | |
| Incomplete | 8/141 | 5.7 (2.8–11.0) | 2/68 | 2.9 (0.7–11.2) | 3/45 | 6.7 (2.1–19.0) | |

[a]One out of 2 samples with G449C was also accompanied by K189T; both cases were polyclonal infections.
[b]Note that the *ubp1* haplotype with R1133S and E1011K was frequently observed with additional SNP mutations (K764N, K774N, and D777G) and a large insert (EQKY) between amino acid positions 2826 and 2827. *dhps* double mutant indicates the A437G A581G mutant; *dhps* triple mutant indicates the A437G A581G K540E mutant. wt, wild type; CI, confidence interval.

majority of parasite lineages harbored sextuple *dhfr/dhps* mutants, *crt* SVMNT, *mdr1* NDFCDY, and *hrp2* and/or *hrp3* deletions. Parasites with this *dhps/dhfr/crt/mdr1* resistance haplotype and both *hrp2* and *hrp3* deleted were first reported in a few cases in Peru in 2006 (61) and were called the $B_{V1}$ lineage. Here, we observed many parasites with the same "$B_{V1}$-type" resistance haplotype although not always with *hrp2* and *hrp3* deletions. With the higher resolution of the Pf AmpliSeq assay, we observed many related lineages with this resistance haplotype that differed in their barcode genotypes, the presence of *hrp2* and *hrp3* deletions, and *coronin* SNPs (Table 2). This indicates gene exchange between lineages, in contrast to the idea of isolated clonal lineages in Peru.

The $B_{V1}$-type lineages (lineage 155 and others) seen after 2008 are most closely related to earlier lineages 158, 141, and 63 in our study samples (Fig. S7). However, the

triple *dhfr* mutants in the $B_{V1}$ lineage are not accompanied by secondary mutations (I164L and the "Bolivian repeat" [BR], a silent 5-amino-acid insertion before codon 30), indicating that they did not emerge from the *dhfr* double mutant lineages circulating in earlier years. In conclusion, the combination of different markers in the Pf AmpliSeq assay allowed the identification of multiple recombining *P. falciparum* lineages in Peru that share increasing SP and CQ resistance, as well as RDT failure, in more recent years.

## DISCUSSION

We designed a new highly multiplexed sequencing assay (Pf AmpliSeq) targeting 13 antimalarial resistance genes, combined with a country-specific 28-SNP barcode for population genetic analysis in Peru and *hrp2* and *hrp3* genes to detect deletions that cause false-negative RDT results. The novelty and strength of this assay are in the many different types of markers multiplexed in one easy workflow, making it suitable for many surveillance use cases. Compared to other amplicon sequencing assays (30, 34, 67) and genetic surveillance platforms (42), the Pf AmpliSeq assay includes the highest number of drug resistance-associated genes combined with an in-country SNP barcode as well as diagnostic resistance markers.

In many remote areas, sample collection for malaria molecular surveillance is possible with DBSs only, which are easily transported at room temperature to a central laboratory for further processing and ideally stored at $-20°C$ to maintain DNA integrity. This assay performs well on DBS samples with parasite densities of $\geq 60$ p/$\mu$L and allows high-accuracy sequencing at a higher depth of coverage (100- to 1,000-fold versus 50- to 100-fold) and a lower cost (see Table S14 in the supplemental material) than WGS. At densities of $<60$ p/$\mu$L, sWGA prior to the Pf AmpliSeq assay increases the number of reads but also increases the error rate and cost. In addition, the Pf AmpliSeq assay is less sensitive to DNA degradation than methods targeting long amplicons.

Initial raw data processing can be performed automatically with the software of the sequencer using the manifest file available at bio-protocol. However, here, we used a Linux-based pipeline that gave us more flexibility and insight during the validation process. Subsequent variant analysis can be performed with minimal bioinformatic skills using any software that can analyze tabulated data. An automated analysis environment would be of great benefit for the rapid output of genetic reports and easy interpretation, increasing the actionable use of data to enable programmatic decision-making.

Only nonvalidated *K13* SNPs were detected at low frequencies in the Peruvian samples, suggesting that ART resistance has not reached Peru, at least in recent years (2003 to 2018). Mutations in other genes in the endocytosis pathway have been associated with decreased sensitivity to ART in *in vitro* studies, and changes in *ubp1* haplotypes paralleled the emergence of *K13* markers in Thailand (60, 68–76). However, the observed *ubp1* and *coronin* mutations here are uncharacterized. Genetic variants in KIC7, Eps15, and Formin2 associated with *in vitro* ART resistance (70–72, 74, 77) were not yet reported when the assay design was being completed. These genes will be included in future versions of the assay.

One main advantage of the Pf AmpliSeq assay is that the panel of targeted amplicons can be easily adapted either to update the panel with newly identified drug resistance-associated genes or other makers of interest or to adapt the assay to a different country or region by exchanging the SNP barcode for in-country resolution. The assay in its current form is limited to South America.

This is the first proof of principle of an NGS amplicon assay targeting *hrp2* and *hrp3* deletions. From the analyzed study samples, 56.3% had both *hrp2* and *hrp3* deleted, but this might be an underestimation since infections containing polyclonal strains with and without deletions will be classified as RDT detectable. For *hrp2*, 22% of samples could not be classified due to conflicting results between *hrp2* amplicons. The difference in read depths among *hrp2* amplicons suggests that only part of the gene is deleted, with part of the first exon still being present. As was recently done in Ethiopia (78), a better characterization of the genomic structure of *hrp2* deletions in Peru will

improve amplicon design. In addition, copy number calling tools could be explored for more systematic classifications of *hrp2* and *hrp3* deletions.

With the combination of phenotypic markers and the 28-SNP barcode in the Pf AmpliSeq assay, we were able to detect changing allele frequencies in resistance-associated genes, but we were also able to identify temporal genetic differentiation in the parasite population, alongside a decrease in genetic diversity. The barcode served to distinguish different lineages in multiple resistance haplotypes (in *dhfr, dhps, crt* and *mdr1*) circulating in recent years and enabled the investigation of drug resistance evolution.

SP and CQ resistance in South America was reportedly spread from a single origin in the lower Amazon (45), with five distinct clonal lineages circulating in Peru before 2006 (19, 61, 79), followed by frequent outcrossing and lineage evolution (79). With the Pf AmpliSeq assay, we observe the same SP/CQ-resistant lineages and their evolution over time, with increased resolution and additional variants detected in other genes.

After 2008, the parasite population in Peru was increasingly dominated by $B_{V1}$-type lineages, which are highly SP/CQ resistant and can frequently escape HRP2-based RDTs (HRP2-based RDTs are not commonly used in Peru). The $B_{V1}$-type lineages seem to be sensitive to ART-MQ (absence of *K13* variants and the *mdr1* CNV), while the high level of CQR (in *crt* and *mdr1*) could be the result of CQ treatment of coendemic *P. vivax* cases. Coinfections with very low *P. falciparum* densities (80) are frequently misdiagnosed and treated as *P. vivax* monoinfection (79). When the $B_{V1}$-type lineages first appeared (2011), we observed strong population differentiation and the loss of secondary *dhfr* mutations, characteristic of Peruvian and Bolivian SP-resistant parasites (45), which supports suggestions from others that this lineage was introduced from other parts of the Amazon (81, 82), possibly Colombia (20, 21) or Suriname (49). Analysis of samples from neighboring countries could shed light on the evolution and spread of resistance and *hrp2* and *hrp3* deletions in the region; however, genomes from South America are heavily underrepresented in publicly available genomic databases.

The COI results varied depending on the analysis method used, and it was difficult to determine which method approached the true COI. The variant calling tool used here assumes a diploid genotype with balanced within-sample allele frequencies, which is frequently not the case in malaria complex infections, and this can complicate COI analyses. Alternative haplotype-based analysis approaches (e.g., SeekDeep [83], HaplotypR [57], and DADA2 [84]) might have a greater ability to unravel complex infections. These tools use a different error-handling approach and thereby can resolve more fine-scale variation and are more suitable for high-resolution relatedness analysis (in combination with identity by descent [IBD] [85]) but are less suitable for confidently calling drug-resistant variants. However, applying these methods to the Pf AmpliSeq data is complicated by the overlapping amplicons in drug resistance genes and requires further validation. While it is possible to perform IBD analysis with the AmpliSeq data, in this study with sampling by convenience, it was deemed less appropriate.

Half of the SNPs in the barcode became fixed in the most recent years, reflecting a true decrease in diversity (as the sample size was sufficient) with closely related clonal lineages. Although our study sample size is comparable to those of previous reports from Peru (e.g., 220 samples in 1999 [79], 104 samples from 1999, and 62 samples from 2006 to 2007 [61]), it represents a limited proportion of the Peruvian *P. falciparum* population. In the future, a more systematic sampling approach, ideally performed at regular times and covering all regions where malaria is endemic, would increase genetic surveillance intelligence to inform malaria control and elimination strategies in Peru.

**Concluding remarks.** We have demonstrated the validity of the Pf AmpliSeq assay for several use cases in malaria molecular surveillance in Peru. With this tool, we have identified *P. falciparum* lineages with an increasing accumulation of drug and diagnostic resistance-associated mutations that can become a serious threat to *P. falciparum* elimination in the Amazon region. The Pf AmpliSeq assay also has the potential to perform well to characterize malaria outbreaks in Peru (81, 86) and predict the origin of imported infections (40, 41, 87).

In Peru, we have established a molecular surveillance network (GENMAL) with the aims of sharing tools, samples, and data; coordinating malaria research efforts; and stimulating dialogue among malaria stakeholders in the region. The network promotes the implementation and use of NGS tools to inform decision-making for malaria elimination and is a platform to support the efficient translation of results into policy and practice. The Pf AmpliSeq assay can be adapted to serve malaria molecular surveillance in the Amazon region and South America, filling the void of genetic data from this region.

## MATERIALS AND METHODS

**Study sites, samples, and controls.** *P. falciparum* qPCR-positive samples (*n* = 312) from previous studies were selected based on geographical representation (Fig. 2) and parasite density (≥100 p/$\mu$L by qPCR). Samples from 2003 to 2017 (*n* = 269) were from published (18, 88) and ongoing studies led by the Universidad Peruana Cayetano Heredia (UPCH) and U.S. Naval Medical Research Unit 6 (NAMRU-6) (*n* = 63). Samples from 2018 (*n* = 5) were *P. falciparum*-positive samples collected in Loreto in a collaboration between the Zero Malaria Programme and UPCH.

DNA from all samples was newly extracted from DBSs using the E.Z.N.A. blood DNA minikit (Omega Bio-Tek, GA, USA) according to the manufacturer's instructions. Per sample, two pieces of ~0.5 cm² were used, and DNA was eluted in a final volume of 100 $\mu$L. *P. falciparum* parasitemia was quantified by qPCR (89). The median parasite density was lower in 2014 to 2018 than in earlier periods (geometric mean density of 2,953 [adjusted *P* value {$P_{adj}$} of 0.0087 by a pairwise *t* test with Benjamini-Hochberg corrections for multiple testing {90}]).

The studies were approved by local ethical review boards. Protocols were registered in the Decentralized System of Information and Follow-Up to Research (SIDISI) (numbers 52707, 61703, 101645, 66235, and 102725), and one clinical trial was registered at ClinicalTrials.gov (identifier NCT00373607). The NAMRU-6 study was approved by its Institutional Review Board in compliance with all applicable federal regulations governing the protection of human subjects (protocol NMRCD.2007.0004). Individuals were included in this study only if signed informed consent included a future-use clause, and secondary use was approved by the Institutional Review Board of the Institute of Tropical Medicine Antwerp (reference number 1417/20).

*P. falciparum* laboratory isolates from cultures (91) and uninfected human blood spotted onto DBS cards were included as controls (details are listed in Table S15 in the supplemental material). Control samples (*n* = 6) with known drug resistance-associated variants from a previous study in Vietnam (92) and genotyped using the SpotMalaria pipeline and WGS were also included. DNA from these controls (100 $\mu$L or 3 DBS punches) was extracted using a QIAamp DNA minikit (Qiagen) according to the manufacturer's instructions. sWGA was performed as previously described (93). Genotypes of the laboratory strains were reported previously (66, 94–96).

**SNP barcode design.** A barcode of 28 biallelic SNPs was designed as described in detail in the supplemental material. Briefly, MalariaGEN *Plasmodium falciparum* Community Project data (97) were used to select SNPs (0.35 ≤ MAF ≤ 0.5) contributing to between-country differentiation using DAPC (98). Two SNPs per chromosome were selected with priority for synonymous SNPs with low pairwise LD.

**Pf AmpliSeq workflow and sequencing.** Library preparation was performed using the AmpliSeq library plus for Illumina kit (Illumina), AmpliSeq custom panels (Data Set S2), and AmpliSeq CD Indexes (Illumina) according to the manufacturer's instructions. The DNA concentrations for the controls and 25 samples were determined using the Qubit v3 high-sensitivity DNA kit (Invitrogen). Controls were diluted to 1 ng input DNA to mimic DBS samples. Study samples (mean DNA concentration of 6.1 ± 0.3 ng/$\mu$L) were not diluted prior to library preparation. For each sample, a single library was prepared, except for a subset of samples to determine reproducibility. Target regions were amplified from 7 $\mu$L DNA (1 to 150 ng) with adjusted cycling conditions (1 cycle of 99°C for 2 min, 21 cycles of 99°C for 15 s, and 1 cycle of 60°C for 8 min) in two reactions and subsequently combined for final library preparation according to the manufacturer's guidelines. Sample libraries were quantified using the Kapa library quantification kit for Illumina platforms (Kapa Biosystems), diluted to 2 nM with low-Tris-EDTA buffer, and pooled to generate a final pool containing 2 nM each library. The denatured library pool (diluted to 18 pM) was loaded onto a MiSeq system (Illumina) for 2 × 300-bp paired-end sequencing (MiSeq reagent kit v3; Illumina). A detailed protocol is available at https://doi.org/10.21769/BioProtoc.4621. A 20% PhiX spike-in (Illumina) was used in one sequencing run to determine the effect on the sequencing quality and the trade-off in coverage. The generated demultiplexed FASTQ files were processed with an in-house analysis pipeline on a Unix operating system computer as described in more detail in the supplemental material.

For the final analysis, we selected 254/312 (81%) samples with good-quality data (<50% missing genotype calls and mean coverage of >15) and retained only one library of replicates (with the lowest missingness). A MAF of 0.01 with a precision of 0.02 (2%) can be detected with sample size of 96 samples, while a MAF of 0.01 with 1% precision can be detected (as determined with https://epitools.ausvet .com.au/oneproportion).

**Detection of *mdr1* and *pm2* copy number variations by PCR.** CNVs in *pm2* and *mdr1* were determined for a subset of samples (*n* = 78) selected randomly from all time periods and districts using qPCR (92). Samples and controls were tested in triplicate, with 20% of the samples being retested for reproducibility and 100% of the results being in agreement.

**Analysis of the complexity of infection.** The COI was estimated using Real McCOIL categorical and proportional methods (99) in R using the following subsets of variants: (i) all biallelic SNPs, (ii) 25 barcode SNPs, and (iii) "core variants," biallelic variants excluding repetitive regions (i.e., *hrp2*, *hrp3*, and MSs) and

mitochondrial and apicoplast regions. Default settings were used for a fixed error; for the categorical method, an upper bound of 10 and an initial COI of 5 were used, and for the proportional method, an upper bound of 15 and an initial COI of 7 were used. We allowed 50% missingness for biallelic SNPs, 20% for the barcode, and 40% for core variants.

We also estimated the proportions of single- and multiple-clone infections based on the number of heterozygous genotypes in (i) the 28-SNP barcode, (ii) biallelic SNPs in *ama1*, and (iii) core variants. Samples with $\geq 1$ heterozygous genotype were considered to contain multiple clones (COI $\geq 2$). Finally, we estimated the COI from the MS-amplified regions. As the different methods generated divergent results, the mode (most frequent value for single versus multiple clones) was determined as the final measure for each sample.

**Validation of detection of *hrp2* and *hrp3* deletions.** The performance of Pf AmpliSeq for *hrp2* and *hrp3* deletion detection was assessed using previous PCR-based *hrp2* and *hrp3* genotypes of 10 included samples (31). The presence or absence of both genes was determined using the mean read depth of *hrp2* and *hrp3* amplicons, as explained in more detail in the supplemental material. Due to the repetitive nature and homologies between the *hrp2* and *hrp3* genes, we used a conservative cutoff value, resulting in a "gray zone" where deletion/presence was left inconclusive when the majority of amplicons were not in accordance (Fig. S9). A final variable for RDT failure (absence of both *hrp2* and *hrp3*) versus RDT detectable (presence of *hrp2* and/or *hrp3*) was created, allowing the classification of samples that were inconclusive for one of the two genes if the other gene was present.

**Population genetic analysis.** Population genetic analyses were performed using the 28-SNP barcode, excluding samples missing >7 SNPs (25%), unless specified otherwise. Samples with heterozygous genotypes were included, justified by the little genetic differentiation ($F_{ST} = 0.05$) between samples with single-clone barcodes and samples with heterozygous barcode genotypes. Genetic differentiation was measured as $F_{ST}$ (100), $G'_{ST}$ (101), and Jost's $D$ (102) using 1,000 bootstraps with the R package diveRsity (103). PCA was performed on the genotype matrix of within-sample allele frequencies using the prcomp function (R stats package v4.0.5) using core variants. Prior to PCA, missing genotypes were replaced by the mean allelic frequency at a locus in all samples. The expected heterozygosity ($H_e$) was calculated (adegenet package [104, 105] in R) using diploid barcode genotypes. A Wilcoxon signed-rank test with continuity correction was used to compare the mean observed heterozygosity ($H_{obs}$) to the $H_e$. LD was measured as the standardized index of association ($\bar{r}D$) in a multilocus analysis using 999 resamplings (method, permutation of alleles), using the poppr package v2.8.6. (106) in R. ML lineages were defined by grouping isolates with similar barcodes using the Hamming distance and clustered based on the maximum distance (farthest neighbor) using poppr. DAPC (98) was performed with cross-validation, and the associated allele loadings for the first four components were determined using the adegenet package in R.

**Analysis of genetic variants.** Barcode allele frequencies were calculated from allele depths to reflect population allele frequencies in complex infections, as described in the supplemental material. Haplotypes were created by combining genotypes of variants of interest (listed in Data Set S1). Frequency tables with confidence intervals were created with the freqtables R package.

**Data availability.** Sample metadata, drug resistance and *hrp2* and *hrp3* haplotypes, barcodes, lineages, locations, and dates, etc., are accessible at https://microreact.org/project/aV7RHNCBmG3sJ2rwzchE4k-peru-2003-2018-molecular-surveillance-with-p-falciparum-ampliseq-assay. Raw data (FASTQ files) are available at the SRA under BioProject accession number PRJNA855317, and individual library accession numbers are listed in the supplemental metadata file at Microreact (see the URL mentioned above). A detailed protocol of the library preparation procedures has been deposited at https://doi.org/10.21769/BioProtoc.4621. It is anticipated that this link will be active by 5 March 2023; until that time, all the data will be available from the corresponding author upon request. Variant files (vcf) and scripts are available upon request. All other data are included in the manuscript and supplemental material.

## SUPPLEMENTAL MATERIAL

Supplemental material is available online only.
**SUPPLEMENTAL FILE 1**, XLSX file, 0.3 MB.
**SUPPLEMENTAL FILE 2**, PDF file, 0.8 MB.

## ACKNOWLEDGMENTS

We thank all clinical, microscopy, and field staff who supported the sample collections and all participants for making their material available for malaria studies. We acknowledge the support of the Dirección Regional de Salud de Loreto for the study authorization and sample collection activities in Loreto. The following reagents were obtained through BEI Resources, NIAID, NIH: *P. falciparum* strain Dd2 (catalogue number MRA-150), contributed by David Walliker; *P. falciparum* strain Dd2_R539T (catalogue number MRA-1255) and *P. falciparum* strain CamWT_C580Y (catalogue number MRA-1251), contributed by David A. Fidock; and *P. falciparum* strain IPC 4912 (catalogue number MRA-1241), contributed by Didier Ménard.

This work was funded by the Belgium Development Cooperation (DGD) under the Framework Agreement Program between the DGD and ITM (FA4 Peru, 2017–2021), and

the sample collections in 2018 were supported by VLIR-UOS (project PE2018TEA470A102; University of Antwerp). Funding for the sample collections led by U.S. Naval Medical Research Unit 6 (NAMRU-6) in 2011 and 2012 was provided by the Armed Forces Health Surveillance Division (AFHSD) and its Global Emerging Infections Surveillance and Response (GEIS) section (P0144_20_N6_01, 2020–2021). The funding agencies had no role in the design of the study.

We declare that we have no competing interests.

Some authors of the manuscript are military service members and employees of the U.S. Government. This work was prepared as part of their official duties. The views expressed in this article are those of the authors and do not necessarily reflect the official policy or position of the Department of the Navy, Department of Defense, or the U.S. Government.

Conception of ideas for the study, A.R.-U., D.G., and J.H.K.; selection of targets and design of the assay, E.R.-V., J.H.K., and N.J.V.D.; sample collections and coordination of field work, C.D.-R., C.F.-M., D.G., H.O.V., and J.-P.V.G.; laboratory experiments, C.F.-M., J.H.K., N.J.V.D., and P.G.; bioinformatics and data analysis, J.H.K., L.L.A., N.J.V.D., and P.M.; writing the first draft of the manuscript, A.R.-U., J.H.K., C.F.-M., and N.J.V.D. All authors reviewed and contributed to the final version of the manuscript.

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
