## [Reviewer comments · Microbiology Spectrum]

Microbiology Spectrum

Malaria molecular surveillance in the Peruvian Amazon with a novel highly multiplexed *Plasmodium falciparum* Ampliseq assay

Johanna Kattenberg, Carlos Fernandez-Miñope, Norbert van Dijk, Lidia Llacsahuanga Allcca, Pieter Guetens, Hugo Valdivia, Jean-Pierre Van geertruyden, Eduard Rovira-Vallbona, Pieter Monsieurs, Christopher Delgado-Ratto, Dionicia Gamboa, and Anna Rosanas-Urgell

Corresponding Author(s): Johanna Kattenberg, Institute of Tropical Medicine Antwerp

Review Timeline:

Submission Date:	March 17, 2022
Editorial Decision:	June 23, 2022
Revision Received:	July 19, 2022
Accepted:	August 2, 2022

Editor: Gemma Moncunill

Reviewer(s): Disclosure of reviewer identity is with reference to reviewer comments included in decision letter(s). The following individuals involved in review of your submission have agreed to reveal their identity: Andres Aranda-Diaz (Reviewer #2)

Transaction Report:

DOI: <https://doi.org/10.1128/spectrum.00960-22>

June 23, 2022

Dr. Johanna Helena Kattenberg
Institute of Tropical Medicine Antwerp
Antwerp
Belgium

Re: Spectrum00960-22 (Malaria molecular surveillance in the Peruvian Amazon with a novel highly multiplexed Plasmodium falciparum Ampliseq assay)

Dear Dr. Johanna Helena Kattenberg:

Thank you for submitting your manuscript to Microbiology Spectrum. Your manuscript has been assessed by two reviewers and they have raised a number of points that we believe would improve the manuscript and may allow a revised version to be published in Microbiology Spectrum. Their reports are below.

Of particular note, raw data (fastq), variant files (vcf) and scripts are only available upon request. Microbiology Spectrum require that authors make data used in publications openly available. A data availability paragraph in the manuscript is required and should include a description of the data along with the repository and accession numbers. For genomic data, accession numbers should be provided for all raw data (SRA records) in addition to any processed data such as genome assemblies (genome or nucleotide records). Methods to generate, process, and analyze the data should be fully described.

Link Not Available

Below you will find also instructions from the Microbiology Spectrum editorial office.

Sincerely,

Gemma Moncunill

Journals Department
Reviewer comments:

Reviewer #1 (Comments for the Author):

This is a very interesting method, particularly in the context of monitoring SNPs and the hrp2 and hrp3 deletions. The population

genetic analyses were less convincing simply because they choose a PCA and FST rather than methods such as IBD. Perhaps the authors could elaborate on why they choose such analysis in this context and no others. It seems that the logic was first to present the method, which seems reasonable. The bioinformatic aspects also need to be addressed considering that this method would likely be used in countries with limited resources. I assume that it is included in the detailed protocol that will become available.

Reviewer #2 (Comments for the Author):

Overall, the manuscript does a good job of describing the design, performance and applications of a multiplexed amplicon sequencing panel for malaria genomics.

As expressed in the manuscript, genotyping tools such as the Pf AmpliSeq assay can augment the data generated by classical epidemiological studies and surveillance.

The manuscript would benefit from clarifications that will improve its readability, as well as a full understanding of the methodologies. The following comments do not question the validity of the results.

My major concern is the absence of a discussion of how experimental choices can influence some of the results. For example, parasitemia and quality of DNA (which should decrease over storage time) can affect detection of variants. The claim that COI increases over time could be explained by those covariates and including relationships (or absence thereof) between parasitemia and measures such as COI would strengthen the claim. This is specially important given that false negatives are high (Table S6, in which the authors could also state what parasitemia was used as a point of reference for the rest of the samples). While false positives seem to be low (Lines 198-201) it would be good to include a discussion about the choice of bioinformatic tools and how they handle errors (with respect to other tools made for amplicon sequencing such as DADA2 or SeekDeep).

For example, in Lines 290-291: are those SNPs observed in samples with higher parasitemia? Are the SNPs observed as minor alleles? Are those newer samples? (Also, Table 3 does not contain information about those SNPs)

Were all samples run in replicate, and if so, were all SNPs in drug resistance-associated markers concordant between replicates?

In the SNP Barcode selection section, the authors state that the selected SNPs "were not under selective pressure from parasite environmental factors, like drug resistance or host immunity". Please clarify how this is defined or reference the studies that support that claim.

There is no explanation of how primers were designed or why some regions "failed in primer design".

Primer specificity was addressed experimentally with uninfected human blood samples. What about other Plasmodium species? Were primers selected for high specificity with other Plasmodium species or the human genome?

Sequencing methods could be clarified. What's the input mass of DNA in the assay? As stated, an absolute value of 1 ng was used for controls (uninfected blood?) but without volumes it's unclear the total input for samples. Were the libraries pooled according to their concentration or in an equivolume way? How many samples were pooled for each MiSeq run? All those details are important to interpret sample and amplicon coverage, DP, etc.

Methods for DNA preparation from laboratory isolate controls is unclear. Are these DBS prepared with laboratory-grown strains (if so, references or details are missing), or are these DNA purchased from a repository?

Link to the detailed protocol for sequencing is missing. Ideally, that protocol contains details on how different primer pools were utilized.

PCA was performed on the genotype matrix (what is this matrix? Are elements the presence or absence of a given allele?). Please clarify

Line 99: more than 2 use cases can be tackled with existing tools (e.g. connectivity, transmission intensity, importation, foci detection, etc with a tool containing only SNPs). Pf AmpliSeq broadens the scope.

Are the numbers reported in line 155 statistics for target regions within a sample or across all samples? I'm actually confused about what target region means here. Is it the aggregate of all targeted regions or is it each of the targets? I don't think it's either because those are numbers reported elsewhere.

The range of reads per sample is big, so a relative measure such as % of total reads within the sample may be more informative than depth of coverage per amplicon.

It's unclear what quality measures are used in lines 157-158. Are these measures of demultiplexed, already filtered reads from BaseSpace or another method that was used to generate the demultiplexed FASTQ files. Or is this part of the in-house analysis pipeline? Did PhiX have an effect on the quality of the run (%Q30, %PF)?

It is unclear how it was determined that there was no contamination in line 186.

Line 191: median DP?

Lines 192-193: Did DP or missingness improve? Also, legend in the graph in Fig S2 is unclear (what axis corresponds to what line/bar?)

Line 204: are all 7.9% of additional genotypes minor alleles within the control samples?

Are the samples mentioned in line 507 not part of the samples enumerated in the first section of the methods?

Lines 242-243: does 'smaller population' refer to the changes in incidence shown in Fig 1? Are those changes observed in each of the regions that were used for this analysis?

Line 284: this legend refers to "isolates" but the analysis was done on samples, not isolated strains, is that right?

It is unclear, from the methods, if the overlapping regions in amplicons that covered drug resistance genes were used to call any variants.

Lines 290-292. Are these mixed infections defined as mixed from SNPs other than the ones in K13? I also personally prefer polyclonal instead of mixed when talking about different clones of *P. falciparum*.

Figure S1: Y axis labels are unintelligible.

Line 376-377: are these discrepancies explained by the breakpoints described in Fekete et al (PMID 34580442)? Can that reference inform the design of amplicons?

Check abbreviation definitions (RTD abbreviation is used in line 59 but defined in line 73, MS is defined in 147 but microsatellites is used before)

Check document for typos. For example, 'scenarios' in line 61, 'resistance' in line 142, 'Target' in line 155, 'became' in line 260, 'appeared' in line 397, spell out 'including' and 'excluding' throughout, 'Pf AmpliSeq' in 204.

145 is missing a reference for previous reports of drug resistance in Peru

Consider use of hyphens for compound words (e.g. hrp2-deleted, resistance-associated)

Define ACT in Figure 1 legend.

In Fig.1 it's unclear when ACT was introduced in the graph.

What does 'always' mean in line 133?

Staff Comments:

Preparing Revision Guidelines

- Point-by-point responses to the issues raised by the reviewers in a file named "Response to Reviewers," NOT IN YOUR COVER LETTER.
- Upload a compare copy of the manuscript (without figures) as a "Marked-Up Manuscript" file.
- Each figure must be uploaded as a separate file, and any multipanel figures must be assembled into one file.

- Manuscript: A .DOC version of the revised manuscript
- Figures: Editable, high-resolution, individual figure files are required at revision, TIFF or EPS files are preferred

Please return the manuscript within 60 days; if you cannot complete the modification within this time period, please contact me. If you do not wish to modify the manuscript and prefer to submit it to another journal, please notify me of your decision immediately so that the manuscript may be formally withdrawn from consideration by Microbiology Spectrum.

Antwerp, June 18th 2022

Dear Editor, dear reviewers,

First of all, we would like to thank the reviewers and editor for their careful review of the manuscript and material and for the comments to help improve the manuscript. Please find below our point-by-point response to the issues raised. We have tried to address all issues and hope that our edits are satisfactory. Line numbers in our responses below refer to the marked-up version.

Response to editor's comments

- Fastq data have now been submitted to SRA under project number PRJNA855317. The data availability paragraph in the manuscript has been adjusted and includes a description of the data along with the repository and project accession number in lines 561-564. Individual library accession numbers are listed in the supplementary file with the study database which is deposited on microreact (link in the manuscript in line 560).

Response to reviewer 1

This is a very interesting method, particularly in the context of monitoring SNPs and the *hrp2* and *hrp3* deletions. The population genetic analyses were less convincing simply because they choose a PCA and F_{ST} rather than methods such as IBD. Perhaps the authors could elaborate on why they choose such analysis in this context and no others. It seems that the logic was first to present the method, which seems reasonable. The bioinformatic aspects also need to be addressed considering that this method would likely be used in countries with limited resources. I assume that it is included in the detailed protocol that will become available.

We would like to thank the reviewer for the comments. We can see that the flow of analysis methods could be clarified, and would like to explain the reasoning behind our selection of methodologies presented.

- The focus of the manuscript is the description and validation of the assay (with samples collected in Peru) to serve several uses cases that can be informative for national malaria control programs. With this aim we first have prioritized analytical methods that are user-friendly and more commonly used. We have explored our data using several methods, however in order to keep the manuscript concise, we present only the most relevant results for the temporal analysis. A more detailed description of several of the methods is presented in Supplementary file 3 to allow replication, and at the same time to stay within the word-limit of the journal.
- Population genetic approaches:
 - Use case transmission intensity
 - We used the PCA analysis as a first step to explore the data. Based on the PCA results our sample set was classified in 3 groups (based on clustering in the PCA presented in supplementary figure S4). These groups were used in subsequent analysis (explained in lines 223-227); therefore, we believe that keeping the supplementary figure and the analysis is needed to clarify how the groups were formed.
 - Subsequently, we calculated the genetic diversity, expressed as expected heterozygosity (H_e), in the 3 different groups, as a proxy of transmission intensity in lines 228-231 and Figure 3A.

- As we observed a large drop in diversity through time, we investigated whether the parasite population in the 3 time periods were genetically differentiated by calculating the F_{ST} (lines 231-234 and Figure 3B). F_{ST} was designed to be used with biallelic variants and is therefore a more appropriate tool than adjusted methods (Jost D, G_{ST}' , etc.) that were developed for multiallelic variants such as microsatellites. Therefore, we prefer to keep this analysis.
 - Use case connectivity of parasite populations
 - We start this use case with the 28-SNP barcode that was specifically designed for the parasite population in Peru. We constructed multi-locus lineages that group the samples based on the 28-SNP barcodes and do a relatedness analysis between the lineages (MSN network presented in Supplementary Figure S7) and investigate the dynamics of these lineages over time (lines 238-246 and Figure 4). Observed lineages do not cluster by geographic area, but change over time (Supplementary Table S11), therefore we investigated the temporal dynamics (Figure 4) rather than the spatial population connectivity. In principle, instead of multilocus-lineage analysis, IBD can be used to infer relatedness between samples, however, generally at least 100 SNPs are used for appropriate resolution in IBD-analysis. Therefore, we prefer to keep the ML-analysis as it more appropriate with the 28 SNPs in the barcode.
 - Next, we use Discriminant analysis of principal components (DAPC), which is similar to PCA but emphasizes differences between predefined populations. This makes DAPC more appropriate to investigate variation between populations rather than a comparison of the variation between individuals as in PCA or IBD. We use DAPC to further explore population structure in the spatial-temporal populations (defined based on the three time periods and district) using all biallelic variants. In addition, the DAPC analysis also identifies the alleles that contribute most to the differentiation observed (lines 247-253), which is essential to identify variants of interest.
- We agree with this reviewer that IBD-analysis can be a powerful tool to perform pairwise comparisons and determine relatedness between samples. This is very useful for the use case of connectivity between areas, investigating transmission chains, or recurrent infection analysis. These use cases were not demonstrated in this study, as the sample collection was not very suitable for these use cases. While we attempted a more systematic sampling approach from a wide range of sites in the country, this was complicated due to the low transmission and heterogeneity of the *P. falciparum* burden in 2018, resulting in only 5 Pf positive samples from 1000s of samples collected. Therefore, samples were included (for convenience) from earlier studies, conducted in different years with a little regional variation, most suitable for a temporal comparison. Our plans are to explore the other use cases with the Pf AmpliSeq in more appropriate sample collections and study sites. We have added the possibility of performing IBD with this data to the discussion lines 413-421.
- We did also explore pairwise IBD (determined using IsoRelate with all biallelic SNPs), which showed similar patterns as those obtained with multilocus lineage analysis and DAPC. The figure below shows a network where edges connecting the samples (dots) represent $\geq 60\%$ IBD (panel A) or $\geq 1\%$ IBD (panel B), using all samples and

controls with sufficient SNPs (<30% missing SNPs). However, we preferred to present the other analyses in the manuscript as explained above. We did not want to overload and complicate the manuscript.

A.

● 2003-2005
● 2008-2012
● 2014-2018
● 9999
● control

B.

● 2003-2005
● 2008-2012
● 2014-2018
● 9999
● control

- We have clarified the different options for variant calling in the bioinformatics pipeline in the discussion (lines 335-342). For someone with bioinformatics skills, our pipeline is easily replicated and follows GATK best practices as much as possible, and is explained in detail in Supplementary file 3. Researchers with fewer skills in linux and in variant-calling pipelines, it is easier to use the programs offered by illumina, either on the local run manager or in the Basespace cloud environment. Manifest files for this latter option are indeed included in the detailed protocol. For clarity to the reviewers this protocol is now added as additional file for the purpose of the review.

- The next step is the analysis and interpretation of the variant data for which we have used different existing R-packages. We are in the process of simplifying the analyses scripts and packaging them in automated pipelines that will generate simple reports and dashboards tuned to the specific requirements and desired use-cases for the end-user. We will present this separately and make these available on github once completed. This has been clarified in the discussion in lines 339-342.

Response to reviewer 2

Overall, the manuscript does a good job of describing the design, performance and applications of a multiplexed amplicon sequencing panel for malaria genomics.

As expressed in the manuscript, genotyping tools such as the Pf AmpliSeq assay can augment the data generated by classical epidemiological studies and surveillance.

The manuscript would benefit from clarifications that will improve its readability, as well as a full understanding of the methodologies. The following comments do not question the validity of the results.

- We appreciate the comments received and have clarified the manuscript and methods throughout the document as much as possible taking the word count restrictions into consideration. Please see highlighted changes in the revised manuscript and supplementary files. A more detailed description of several of the methods is presented in Supplementary file 3 to allow replication, and at the same time to stay within the word-limit of the journal

My major concern is the absence of a discussion of how experimental choices can influence some of the results. For example, parasitemia and quality of DNA (which should decrease over storage time) can affect detection of variants.

- We fully agree with the reviewer that sample quality and thus integrity of the template DNA are important factors in the success of the assay. Therefore new DNA extractions were performed for this study. This was clarified in line 461. Extracted DNA was not stored for a long period before library preparation for the Pf AmpliSeq assay. The DNA was checked with a qPCR and we selected samples with good amplification in the qPCR. As we target short amplicons, this assay is less sensitive to DNA degradation than assays amplifying long targets, and in our experience, successful qPCR-amplification is a good estimator of library preparation success.
- The effect of parasitemia is explored with a 3D7 control isolate in a dilution series and results are presented in lines 180-184 and it is discussed in lines 327-330. This assay performs well on DBS samples with parasite densities ≥ 60 p/ μ l. At densities < 60 p/ μ l, sWGA prior to the Pf AmpliSeq assay increases the number of reads, but also the error rate.
- We do not expect the reported parasite densities to have been impacted by the long storage time. Samples were quantified by qPCR on fresh DNA extracts. All samples included in the Pf AmpliSeq assay had high parasite density (we selected samples with ≥ 100 p/ μ l by qPCR to be sure) as reported in line 456.
- Samples that did not perform well in the library preparation (due to sample quality, quantity, or any other reason) were not included in the analysis. We used a cut-off of a coverage of 15 and missingness of 50%, and samples with values below any of these two

variables were excluded from analysis as reported in lines 505-508. Sequencing depth was not lower in older samples from 2003-2005 vs. newer samples from 2014-2018.

- Sample considerations are also an element taken into account in the detailed protocol, which we now added as additional file for the purpose of the review. In our experience, samples that are amplified in the qPCR with a decent amount of template, usually perform well in the AmpliSeq, irrespective of how long the DBS have been stored. Frequent freeze-thawing of DNA after extraction should be kept to a minimum. For routine surveillance (for which this assay is meant), long term storage and DNA degradation is expected to be even less of a problem, as filter papers will be processed within ~6 months after collection. We have added recommendations for sample collection in the discussion in lines 325-334.

The claim that COI increases over time could be explained by those covariates and including relationships (or absence thereof) between parasitemia and measures such as COI would strengthen the claim. This is specially important given that false negatives are high (Table S6, in which the authors could also state what parasitemia was used as a point of reference for the rest of the samples).

- We thank the reviewer for the suggested explanation, however:
 - We included samples that were amplified and quantified by qPCR past a density of 100 p/ul as reported in line 456.
 - Mean parasite density is lower in more recent years (2014-2018) compared to the earliest years, while COI is higher in those years. This was added to lines 463-465. Therefore there is no association between higher COI with higher parasite density in our study population as suggested by the reviewer.
 - There is not a strong correlation between sequencing depth vs parasitemia in the study samples, as libraries are diluted in an equimolar pool before sequencing. Details for sequencing conditions are now better described in lines 493-501.
 - For the majority of samples parasite density was sufficient for good amplification of targets in the library preparation procedures. Poorly amplified samples below the inclusion threshold, were excluded from analysis as reported in lines 505-508.
 - We agree that the COI determination and observed trends therein could be affected by technical issues, and therefore we do not draw strong conclusions from this analysis. However, we think this is due to the nature of the variant calling process and methods for COI-calling rather than an issue of sample quantity or quality. We have expanded this discussion for further clarification (lines 408-421).

While false positives seem to be low (Lines 198-201) it would be good to include a discussion about the choice of bioinformatic tools and how they handle errors (with respect to other tools made for amplicon sequencing such as DADA2 or SeekDeep).

- We have clarified the section on how tools such as dada2 or seekdeep handle errors compared to our pipeline in lines 413-415.
 - We did not use these other tools as there are some challenges in applying them to the AmpliSeq data. The main issue being that these tools do not work well with the

overlapping design of the amplicons as primer sequences are found within the overlapping regions, i.e. within multiple amplicons. The primer sequences are used to demultiplex the reads by amplicon in SeekDeep, which we tested, and it underperformed when using the entire targeted region. In the amplicon variant calling pipelines from Illumina (local run manager or basespace), the Smith-waterman or burrows-wheeler aligners are used to align AmpliSeq data, so we decided to follow this. Another reason for us to use bwa for alignment is that it simplifies annotation of variants that is important for discovery of novel variants in resistance genes. As we are detecting drug resistant variants we want to be very confident in the variants we reports therefore in addition to the aligners used we hard filter our variants to reduce error as described in the supplementary methods file.

For example, in Lines 290-291: are those SNPs observed in samples with higher parasitemia? Are the SNPs observed as minor alleles? Are those newer samples? (Also, Table 3 does not contain information about those SNPs)

- It has been clarified that the alleles in K13 were always found in as minor alleles in mixed infections in lines 259.
- Only K13 variants that were observed in more than one included sample (i.e. only G449C) are reported in Table 3. This has been clarified in lines 1080-1081.
- These are not very important variants, but we did want to mention the observations in case these variants will be reported more frequently in later years. But since the numbers were so low we did not investigate trends. The absence of validated markers of ART-resistance is the main message of this section.

Were all samples run in replicate, and if so, were all SNPs in drug resistance-associated markers concordant between replicates?

- The samples in general were not run in replicates, only where specified for the analysis of reproducibility. This has been clarified in Lines 493-495.

In the SNP Barcode selection section, the authors state that the selected SNPs "were not under selective pressure from parasite environmental factors, like drug resistance or host immunity". Please clarify how this is defined or reference the studies that support that claim.

- We selected SNPs based on linkage disequilibrium and based on the annotations, prioritizing synonymous SNPs and removing any SNPs from genes that are exposed to the outer membrane as well as any regions near drug resistance associated genes. This is described in more detail in the supplementary file 3.

There is no explanation of how primers were designed or why some regions "failed in primer design".

- Primers were designed in Design Studio (Illumina) by the Illumina concierge team using the 3D7 genome and Malariagen variant database. This is now clarified in the manuscript in lines 137-140. If primers failed the design, this is because in that region the primers and amplicons could not be designed within the constraints of the multiplex assay (e.g. high AT rich sequences, high genetic variability surrounding target regions) as explained in lines 141-144.

Primer specificity was addressed experimentally with uninfected human blood samples. What about other Plasmodium species? Were primers selected for high specificity with other Plasmodium species or the human genome?

- Primer specificity of designed primers was investigated using the primer blast tool and the nucleotide collection of NCBI. Based on this, no cross-reactivity was expected with other human plasmodium species therefore this was not further tested in the lab and we tested only human DNA as this is a primary 'contaminant' present in every sample and we wanted to test whether this could cause unspecific amplification, especially in the absence or in the case of very low template (Pf) DNA.
- In addition, even if there is a co-infection with another species (e.g. *P. vivax*) combined with unspecific amplification, resulting sequences will not align properly to the reference sequence due to dissimilarities in the genomes and will be discarded before variants are called. We have investigated this in more detail for our Pvivax AmpliSeq assay that was performed on *P. falciparum* and *P. knowlesi* DNA (Kattenberg et al. Frontiers in Cellular and Infection Microbiology, in review).

Sequencing methods could be clarified. What's the input mass of DNA in the assay? As stated, an absolute value of 1 ng was used for controls (uninfected blood?) but without volumes it's unclear the total input for samples. Were the libraries pooled according to their concentration or in an equivolume way? How many samples were pooled for each MiSeq run? All those details are important to interpret sample and amplicon coverage, DP, etc.

- Sequencing methods were clarified in lines 493-500 regarding the input amounts for the library preparation and equimolar pooling.
- Recommendations for these steps are also included in the detailed protocol.

Methods for DNA preparation from laboratory isolate controls is unclear. Are these DBS prepared with laboratory-grown strains (if so, references or details are missing), or are these DNA purchased from a repository?

- Methods for culturing and DNA preparation from control laboratory isolates were clarified in lines 477-480. Supplementary table S15 contains a list of the strains used.

Link to the detailed protocol for sequencing is missing. Ideally, that protocol contains details on how different primer pools were utilized.

- The detailed protocol of the procedures is going through revision and the link was not yet available upon first submission of the manuscript. For clarity to the reviewer it is now added as additional file, and the link in the paper will be updated as soon as the link is operational (expected soon). Indeed the protocol contains exact information on the primer sequences, which primer is in which pool and exact procedures for replicating the library preparation.

PCA was performed on the genotype matrix (what is this matrix? Are elements the presence or absence of a given allele?). Please clarify

- We used prcomp in R for the PCA analysis, which requires a matrix as input. This matrix is the within-sample frequency of each allele for each sample. This was clarified in lines 542.

Line 99: more than 2 use cases can be tackled with existing tools (e.g. connectivity, transmission intensity, importation, foci detection, etc with a tool containing only SNPs). Pf AmpliSeq broadens the scope.

- We agree with this comment of the reviewer. The sentence has been changed to 'Currently, there is no multifunctional tool that includes a combination of more than two types of markers (*i.e.* SNP-barcodes, drug resistance, etc.) to serve several use cases.' in lines 98-100.

Are the numbers reported in line 155 statistics for target regions within a sample or across all samples? I'm actually confused about what target region means here. Is it the aggregate of all targeted regions or is it each of the targets? I don't think it's either because those are numbers reported elsewhere. The range of reads per sample is big, so a relative measure such as % of total reads within the sample may be more informative than depth of coverage per amplicon.

- This is the overall mean of reads generated per library. This includes all samples (poor and good) and all controls (positive and negative controls). This was clarified in lines 149-156.

It's unclear what quality measures are used in lines 157-158. Are these measures of demultiplexed, already filtered reads from BaseSpace or another method that was used to generate the demultiplexed FASTQ files. Or is this part of the in-house analysis pipeline?

- What is used is the amount of poor quality reads that were trimmed away using trimmomatic in the in-house pipeline. This has been clarified in lines 153-156.
- Reads are automatically demultiplexed by the MiSeq local run manager after the sequencing run.

Did PhiX have an effect on the quality of the run (%Q30, %PF)?

- The manufacturer does not recommend adding PhiX for the AmpliSeq assays in the standard guidelines. However, as PF has such a high AT percentage, we tried to add the PhiX in an attempt to increase nucleotide diversity and thereby also the overall Q30 score.
- The %Q30 was improved. However, we still had lots of trimming of poor quality reads and on top of that lower nr of reads/sample because 20% of the sequencing output is taken up by the PhiX library. So in the end, it is not worth adding 20% PhiX. Overall sequencing quality is better, but per sample, without PhiX and after trimming away reads with poor quality, you end up with more good quality reads without adding PhiX. This is clarified in lines 153-156.
- There is still another benefit of adding PhiX, which allows better troubleshooting of a run if needed. This is described in the detailed protocol (we add 1-5% PhiX).

It is unclear how it was determined that there was no contamination in line 186.

- Contamination is defined as the presence of *P. falciparum* sequences (which should not be there in a human uninfected control). The sentence was clarified in lines 173-178.

Line 191: median DP? -> corrected

Lines 192-193: Did DP or missingness improve? Also, legend in the graph in Fig S2 is unclear (what axis corresponds to what line/bar?)

- DP improved especially, this is now added.

- The caption in figure S2 was adjusted to clarify the bars and lines.

Line 204: are all 7.9% of additional genotypes minor alleles within the control samples?

- Most of them yes. In one case, for the K13 genotype its more balanced (mixed C580Y & I543T), however the comparison assay (WGS) had much lower depth than the AmpliSeq.

Are the samples mentioned in line 507 not part of the samples enumerated in the first section of the methods?

- Yes, we used previously generated PCR data from included samples that were described at the start of the methods. This is clarified in lines 527-528

Lines 242-243: does 'smaller population' refer to the changes in incidence shown in Fig 1? Are those changes observed in each of the regions that were used for this analysis?

- No what is meant in the smaller total population size as indicated by Hobs<He in lines 233-234. We added the word 'sized' to clarify the meaning.

Line 284: this legend refers to "isolates" but the analysis was done on samples, not isolated strains, is that right?

- Changed 'isolates' to 'samples' in line 392 and 1131.

It is unclear, from the methods, if the overlapping regions in amplicons that covered drug resistance genes were used to call any variants.

- Yes these were used to call variants as with our applied variant calling methods, we could easily include the overlapping regions. (which is one of the reasons why we used this approach and not others that split up the amplicons).
- This detail was added to the supplementary methods file

Lines 290-292. Are these mixed infections defined as mixed from SNPs other than the ones in K13? I also personally prefer polyclonal instead of mixed when talking about different clones of *P. falciparum*.

- We understand the confusion with 'mixed' infections that commonly refers to mixed species infections. In this case we indeed mean polyclonal infections and this was changed in the manuscript in lines 259, 359, Table 2 and Table 3

Figure S1: Y axis labels are unintelligible.

- We removed the axis labels in figure S1, these were the amplicon IDs.

Line 376-377: are these discrepancies explained by the breakpoints described in Fekete et al (PMID 34580442)? Can that reference inform the design of amplicons?

- Thank you for the suggestion. However, they do not match the breakpoints in Fekete et al. However, this is a very good line of thought. The difference in performance of the amplicons does match with locations of the break points observed in isolates in Peru which is reported in the supplementary data of the malaria gen pf 7 draft manuscript (in preparation). We are in progress of exploring this further and redesigning the ampliseq amplicons and analysis pipeline with this information, but this is work in progress. As this is information from a manuscript in preparation shared with us in confidence, we did not include this in detail in this paper, but instead we had phrased it more generally.

- We have now included a reference to the breakpoints observed in Ethiopia in Feleke et al. in lines 363-365 to make the comparison and to be more clear on the suggestion of the breakpoints.

Check abbreviation definitions (RTD abbreviation is used in line 59 but defined in line 73, MS is defined in 147 but microsatellites is used before)

- Corrections were made, please see track changes in manuscript document. Thank you for bringing it to our attention.

Check document for typos. For example, 'scenarios' in line 61, 'resistance' in line 142, 'Target' in line 155, 'became' in line 260, 'appeared' in line 397, spell out 'including' and 'excluding' throughout, 'Pf AmpliSeq' in 204.

- Corrections were made, please see track changes in manuscript document. Thank you for bringing it to our attention.

145 is missing a reference for previous reports of drug resistance in Peru

- Previous reports of drug resistance in Peru are referenced in lines 119-121 (references 51-54) and all targeted genes are referenced per gene in Table 1.

Consider use of hyphens for compound words (e.g. hrp2-deleted, resistance-associated)

- Corrections were made, please see track changes in manuscript document. Thank you for bringing it to our attention.

Define ACT in Figure 1 legend.

- Corrections were made, please see track changes in manuscript document. Thank you for bringing it to our attention.

In Fig.1 it's unclear when ACT was introduced in the graph.

- Corrections were made, please see the adjusted figure and the legend was clarified in line 1088.

What does 'always' mean in line 133?

- We rephrased the sentence to 'was the recommended first-line treatment from the start...' in lines 1089-1091

August 2, 2022

Dr. Johanna Helena Kattenberg
Institute of Tropical Medicine Antwerp
Antwerp
Belgium

Re: Spectrum00960-22R1 (Malaria molecular surveillance in the Peruvian Amazon with a novel highly multiplexed Plasmodium falciparum Ampliseq assay)

Dear Dr. Johanna Helena Kattenberg:

Your manuscript has been accepted, and I am forwarding it to the ASM Journals Department for publication. You will be notified when your proofs are ready to be viewed.

Sincerely,

Gemma Moncunill
Editor, Microbiology Spectrum

Journals Department
Supplemental Material: Accept
Supplemental Dataset 1: Accept